# LEAFY is a pioneer transcription factor and licenses cell reprogramming to floral fate

Run Jin [1], Samantha Klasfeld[1], Yang Zhu [1], Meilin Fernandez Garcia [2,3], Jun Xiao[1,4,5], Soon-Ki Han [1,6], Adam Konkol [1] & Doris Wagner [1✉]

Master transcription factors reprogram cell fate in multicellular eukaryotes. Pioneer transcription factors have prominent roles in this process because of their ability to contact their cognate binding motifs in closed chromatin. Reprogramming is pervasive in plants, whose development is plastic and tuned by the environment, yet little is known about pioneer transcription factors in this kingdom. Here, we show that the master transcription factor LEAFY (LFY), which promotes floral fate through upregulation of the floral commitment factor *APETALA1* (*AP1*), is a pioneer transcription factor. In vitro, LFY binds to the endogenous *AP1* target locus DNA assembled into a nucleosome. In vivo, LFY associates with nucleosome occupied binding sites at the majority of its target loci, including *AP1*. Upon binding, LFY 'unlocks' chromatin locally by displacing the H1 linker histone and by recruiting SWI/SNF chromatin remodelers, but broad changes in chromatin accessibility occur later. Our study provides a mechanistic framework for patterning of inflorescence architecture and uncovers striking similarities between LFY and animal pioneer transcription factor.

[1] Biology Department, University of Pennsylvania, Philadelphia, PA 19104-5157, USA. [2] School of Medicine, Philadelphia, PA 19104-5157, USA. [3] Present address: Icahn School of Medicine at Mount Sinai, New York City, NY 10029-5674, USA. [4] Present address: Institute of Genetics and Developmental Biology, Chinese Academy of Sciences, Beijing 100190, China. [5] Present address: Centre of Excellence for Plant and Microbial Science (CEPAMS), The John Innes Centre, Norwich Research Park, Norwich NR4 7UH, UK. [6] Present address: Institute of Transformative Bio-Molecules (WPI-ITbM), Institute for Advanced Research, Nagoya University, Chikusa, Nagoya 464-8601, Japan. ✉email: wagnerdo@sas.upenn.edu

Chromatin prevents expression of inappropriate or detrimental gene expression programs, allowing the formation of distinct cell types from the same genome[1]. The basic unit of chromatin is the nucleosome comprised of 147 base-pairs of DNA wrapped nearly two turns around the histone octamer[2]. Chromatin is further compacted by nucleosome/nucleosome interactions and by the linker histone H1, which associates with the dyad (midpoint) of the nucleosome and the linker DNA near the nucleosome entry and exit[3]. During cell fate reprogramming in eukaryotes, master transcription factors silence and activate new gene expression programs in the context of chromatin[4–6]. While it is easy to imagine how master transcription factors bind to active genes in open chromatin to trigger silencing, it is difficult to envision how these sequence-specific binding proteins can access their cognate motifs in silent chromatin to activate gene expression. This is because nucleosomes are refractory for most transcription factor binding[7–11]. However, a special class of transcription factors, termed pioneer transcription factors, can access their cognate binding motifs in the nucleosome[1,4,12–17]. These pioneer transcription factors play important roles in cell fate reprogramming. For example, the mammalian pioneer transcription factor FoxA reprograms fibroblast to hepatocytes[16,18–20], while the Oct4, Klf4 and Sox2 pioneer transcription factors reprogram fibroblasts to induced pluripotent stem cells[14,21,22]. Defining criteria for pioneer transcription factors are (1) ability to bind cis motifs in the context of a nucleosome both in vitro and in vivo, (2) facilitating access of additional, non-pioneer, transcription factors to target loci via local chromatin opening and (3) cell fate reprograming[13–15,23,24].

Plant development occurs after embryogenesis and is tuned by the environment, a likely adaptation to the sessile lifestyle[25]. Not surprisingly, many master transcription factors that reprogram cell fate have been identified in plants. For example, the bHLH transcription factors MUTE and FAMA reprogram leaf epidermal cells to guard cell fate[26,27], while AP2 family transcription factors WOUND INDUCIBLE1 and BABY BOOM cause dedifferentiation upon wounding and somatic embryo formation on seedlings, respectively[28,29]. The NF-Y complex transcription factor LEAFY COTYLEDON1 (LEC1) also promotes embryo fate in seedlings[30]. The helix-turn-helix (HTH) transcription factor LEAFY (LFY) is necessary and sufficient to trigger flower formation on inflorescences[31,32] and reprograms cells in roots of growing seedlings to flower fate, when ectopically expressed together with the pluripotency factor WUSCHEL[33]. In root explants, inducible activation of LFY is sufficient to trigger synchronous, abundant flower formation bypassing elaboration of a shoot[34]. Finally, MADS box transcription factors of the SEPALLATA (SEP) family can reprogram cauline leaves into floral organs[35].

Pioneering activity has been proposed for several of these plant transcription factors, including LEC1[36]. The NF-YB homolog LEC1[37–39] has not been tested for pioneer activity in plants. In animals, NF-YB is recruited as a component of the histone fold NF-Y complex, which binds closed chromatin by displacing histones[15,40–45]. MADS box transcription factors and LFY associate with genomic regions in Arabidopsis that are not in an open chromatin configuration based on DNAse hypersensitivity[46,47]. While the available data hint at possible pioneer factor activity, it has not yet been established for any of these plant transcription factors whether they can indeed access nucleosome occupied binding sites in vivo or in vitro. The ability of LFY to reprogram root cells to floral fate prompted us to investigate whether LFY acts as a pioneer transcription factor. A key direct target of LFY is APETALA1 (AP1), a MADS box transcription factor that commits primordia in the inflorescence to flower fate[48–50]. LFY upregulates AP1 both directly and indirectly, via a series of coherent 'and' logic

feed-forward loops[48,51,52]. The LFY binding site at the AP1 locus is critical for locus activation[53] and LFY association with this site has been structurally characterized[54].

Here we use complimentary biochemical, genomic and structural approaches to test whether LFY is a bonafide pioneer transcription factor. We find that in vitro LFY binds with high affinity and specificity to a native regulatory fragment from the AP1 locus in the context of a nucleosome. In vivo, the majority of the LFY bound sites, including that at AP1, are nucleosome occupied and isolated LFY-associated DNA fragments are co-bound by histones. LFY displaces linker histone H1 and recruits SWI/SNF chromatin remodelers at the AP1 locus, this triggers subsequent changes in chromatin accessibility. Our findings identify LFY as a pioneer transcription factor, link pioneer activity of LFY to competency for cell fate reprogramming and pave the way for understanding the molecular basis for the developmental plasticity of plants.

## Results

**LFY binds nucleosomes in vitro.** To test ability of the helix-turn-helix transcription factor LFY to occupy its binding site in the context of nucleosomes, we focused on its key target AP1[48]. Analysis of a published MNase-seq dataset[55] revealed a nucleosome whose midpoint (dyad) was positioned near the functionally important[53] LFY binding site at the AP1 locus (Fig. 1a). We cloned and Cy5 labeled a 160 bp DNA fragment that encompasses this nucleosome with the LFY target site at its center. Using recombinant, purified, full-length LFY protein (Supplementary Fig. 1a) we tested LFY binding to this naked AP1 regulatory DNA fragment by electrophoretic mobility shift assays (EMSAs). LFY bound its target site in the AP1 fragment with high affinity based on the apparent dissociation constant ($K_D$), and with high specificity as binding was abolished when we mutated the known consensus motif[56,57] (Supplementary Fig. 1b, d). Using established procedures[14,15], we next assembled the 160 base-pair native AP1 regulatory region, containing the wild-type or a mutated LFY binding site, with purified recombinant histones by salt gradient dilution (Supplementary Fig. 1e, f). Both DNA fragments formed stable nucleosomes, which we further purified using glycerol gradients (Supplementary Fig. 1f, g). EMSAs revealed that LFY also associates with the nucleosomal template with high affinity (Fig. 1b). The LFY target site is near the nucleosome dyad, where it resides in vivo (Fig. 1a) and where histone-DNA interactions are strongest[58,59]. LFY did not bind the nucleosomal substrate when its consensus motif was mutated, indicating that LFY binds the nucleosome in a sequence specific manner (Fig. 1b). Based on the apparent dissociation constants (5–10 nM), the affinity of LFY for the endogenous sequence-based nucleosomal template is high (Fig. 1c), albeit slightly lower than those described for mammalian pioneer transcription factors (1-2 nM or 2–6 nM)[14,15]. Like LFY, these pioneer transcription factors also have higher affinity for the naked DNA than for the nucleosomal template[15]. We conclude that LFY binds its binding motif in the context of a nucleosome in vitro.

We have previously shown that LFY activates AP1 together with a MYB transcription factor termed LATE MERISTEM IDENTITY 2 (LMI2) in a coherent 'and' logic feed-forward loop[51] (Fig. 1d). The LMI2 binding site[51,60,61] is 30 bp away from the LFY binding site near the nucleosome dyad (Fig. 1a). This provides an opportunity to test whether this transcriptional activator of AP1 can also associate with nucleosomes in vitro. After purifying recombinant LMI2 protein, we first probed LMI2 association with its binding site in the 160 bp naked AP1 regulatory DNA in vitro (Supplementary Fig. 1a). LMI2 bound the AP1 locus regulatory DNA in a binding site-specific manner

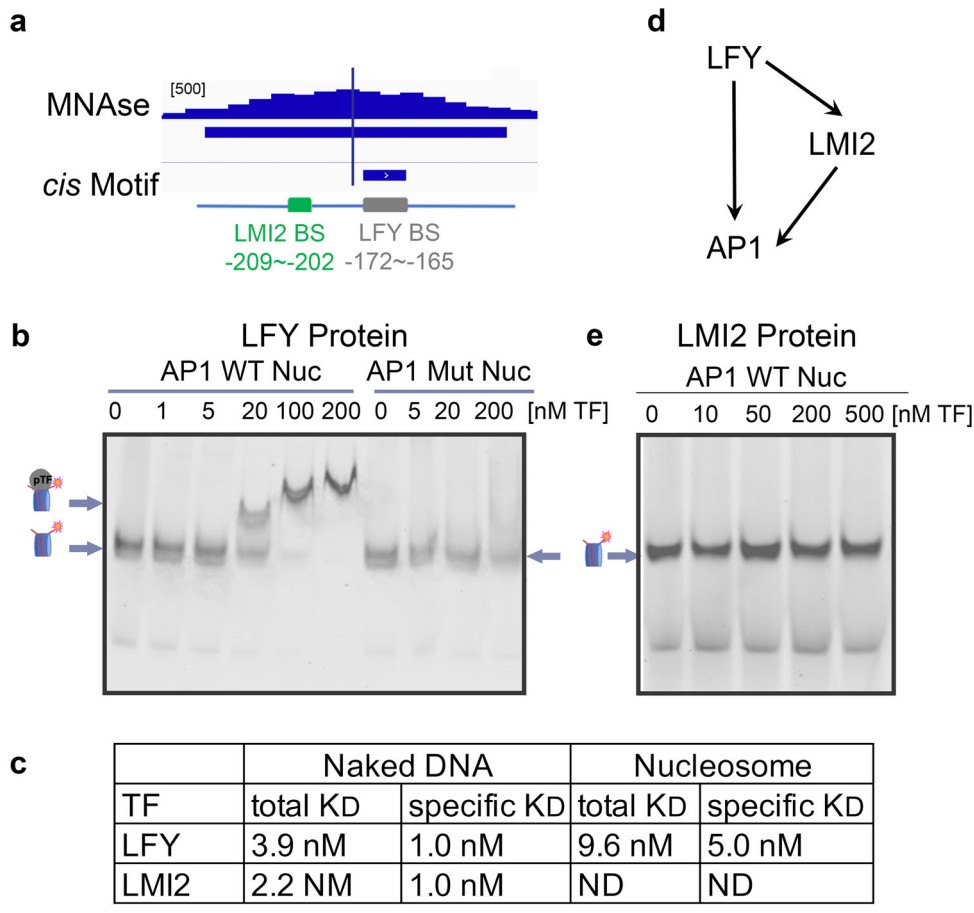

**Fig. 1 The helix-turn-helix transcription factor LFY binds specifically and with high affinity to nucleosomal DNA in vitro. a** Top: Nucleosome at the *AP1* regulatory region positioned over a functionally important LFY binding site[53, 55]. Vertical line: nucleosome mid-point (dyad). Browser screenshot: MNase-seq reads. Below: significant nucleosome (DANPOS *q* value $\leq 10^{-100}$) with color saturation proportional to the signal strength. Short horizontal line: LFY binding motif (consensus core = CCANTGG)[53, 54, 57]. Bottom: *AP1* locus regulatory region with binding site for LFY and for the MYB transcription factor LMI2[51, 60] and their distance from the transcription start site. **b** EMSA of LFY binding to native *AP1* regulatory DNA assembled into nucleosomes containing a wild-type (left) or a mutated (right) binding motif at the nucleosome dyad. Arrows and drawings on the left indicate nucleosome alone (bottom) and transcription factor nucleosome complex (top). The supershift observed at high molar excess is typical of nucleosomal EMSAs[14, 15]. The experiment was repeated three times with similar results. **c** Apparent dissociation constants ($K_D$ in nM) of LFY and LMI2 binding based on the decrement of nucleosome (total binding) or the first bound nucleosome (specific binding) as described in Ref. [14]. ND: not detectable. **d** LFY and LMI2 feed-forward loop for transcriptional activation of *AP1*[51]. **e** EMSA of LMI2 binding to native *AP1* regulatory DNA containing the wild-type binding motif assembled into a nucleosome. The experiment was repeated three times with similar results. See also Supplementary Fig. 1.

and with high affinity (Supplementary Fig. 1c, d; Fig. 1c). We next tested LMI2 association with its binding motif in the context of a nucleosome. Unlike LFY, LMI2 was unable to bind its target sequence in a nucleosome even at high molar excess (Fig. 1e). Thus, although LMI2 in vitro shows higher affinity for the naked *AP1* locus DNA fragment than LFY, its association with the DNA is blocked by a nucleosome.

**LFY binds nucleosomes in vivo.** We wished to use root explants to test whether LFY can associate with nucleosomes in vivo. This system uses inducible (dexamethasone steroid triggered) nuclear entry of a glucocorticoid receptor hormone binding domain fusion of LFY (35S:LFY-GR)[48]. In root explants, unlike in inflorescences, 35S:LFY-GR activation triggers synchronous, abundant, flower formation[34] (Supplementary Fig. 2a, b). We characterized the root explant reprogramming system by examining the kinetics of LFY binding and gene activation, focusing on the *AP1* locus. LFY binding to *AP1* was rapid, with strong occupancy observed already 20 min after dexamethasone application (Supplementary Fig. 2b, c). Robust *AP1* induction was first

observed 24 h after LFY activation with further increases in *AP1* message accumulation until day five (Supplementary Fig. 2d). To probe whether the LFY binding site at the *AP1* locus is nucleosome occupied, we conducted micrococcal nuclease (MNase) digestion followed by tiled oligo qPCR. This uncovered a nucleosome centered over the LFY binding site at the *AP1* locus at a similar position as in the published dataset (Supplementary Fig. 2e and Fig. 1a).

To assess nucleosome occupancy at LFY bound sites genome-wide, we next conducted chromatin immunoprecipitation (ChIP) and MNase digestion in root explants one hour after steroid or mock treatment (Fig. 2a) followed by sequencing. ChIP-seq with LFY-specific antibodies[57] identified 1177 significant (MACS2 summit qvalue $\leq 10^{-10}$) LFY binding peaks in the steroid treated tissue (Supplementary Data 1). The quality of the ChIP-seq data was high based on replicate concordance (Supplementary Fig. 3a–c). Moreover, the majority of the LFY binding peaks contained the known LFY consensus motif[56,57] under the peak summit (Supplementary Fig. 3d, e). Lastly, a significant fraction of the LFY binding peaks identified in root explants overlapped with those previously identified

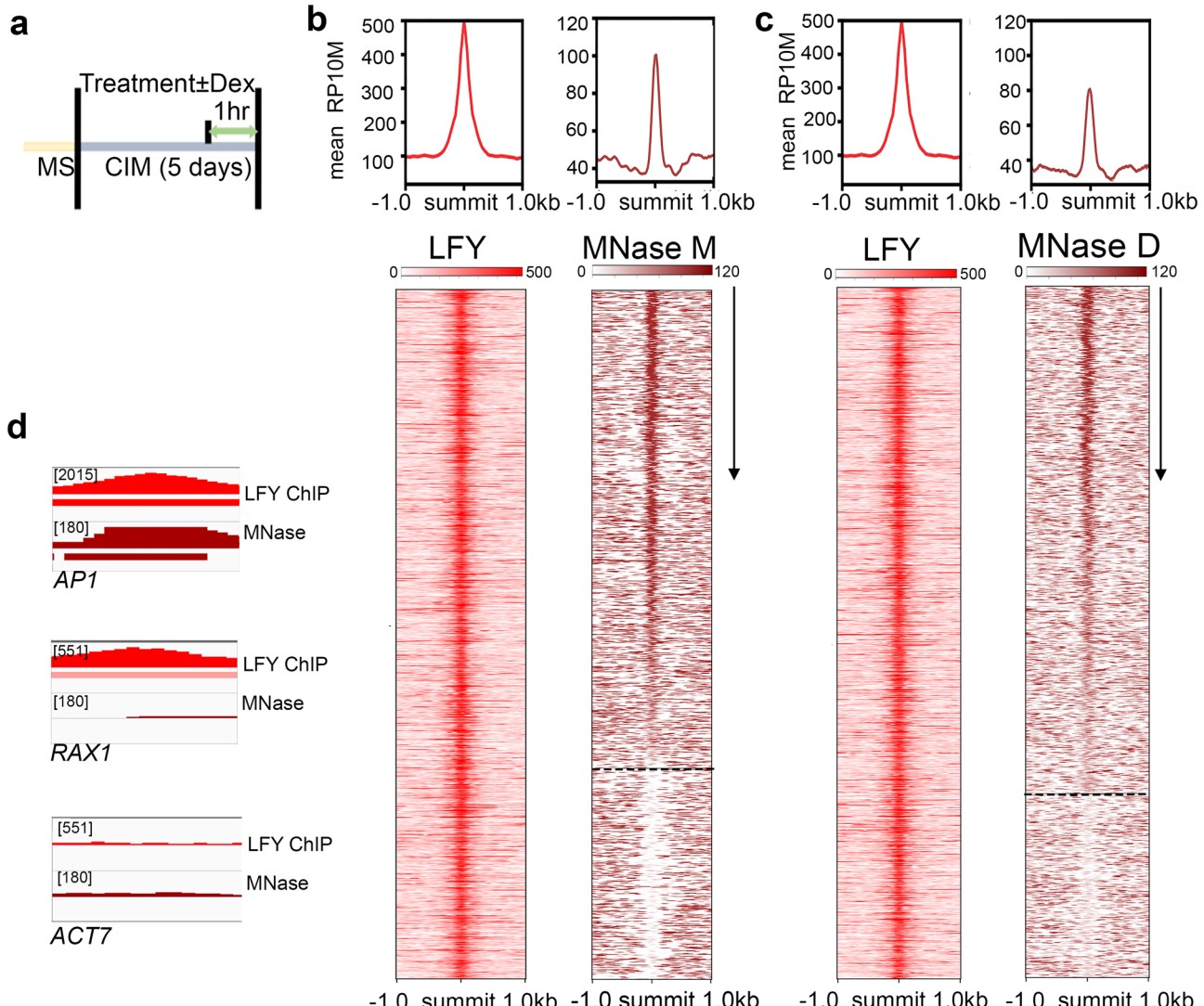

**Fig. 2 LFY binds nucleosomes in vivo during root explant reprogramming. a** Experimental setup to test in vivo association of LFY with nucleosomal DNA and timeline for LFY-GR treatments. MS: seedling growth medium, CIM: root explant medium. **b**, **c** Top: Mean signal of significant LFY ChIP-seq (RP10M; $q$ value $\leq 10^{-10}$, MACS2) and of nucleosome occupancy (DANPOS; $q$ value $\leq 10^{-50}$) in root explants ± 1 kb of significant LFY peak summits. Bottom: ChIP-seq and MNase-seq signal in a 2 kb region centered on significant LFY peak summits and ranked by nucleosome occupancy levels (arrow) in the LFY peak summit region (±75 bp). Root explant treatments: LFY-GR ChIPseq: one-hour steroid treatment. MNase: low MNase digestion of one-hour mock treated LFY-GR samples (b) or low MNase digestion of one-hour steroid treated LFY-GR samples (c). Dotted lines separate nucleosome occupied (top) from nucleosome free (bottom) LFY binding sites. **d** Browser view of LFY ChIPseq and MNase-seq at known LFY targets (*AP1* and *RAX1*) and at the housekeeping gene *ACTIN7* (*ACT7*). Below: significant LFY ChIP-seq peak and nucleosome; color saturation proportional to the negative log 10 $q$ value and the signal strength, respectively. See also Supplementary Figs. 2 - 4.

during the switch to floral fate[57] (Supplementary Fig. 3e). To probe nucleosome occupancy, we employed MNase-seq after using both low digestion, which is customarily used to capture 'fragile' nucleosomes when investigating pioneer transcription binding to nucleosomes in vivo[16], or standard (high) MNase digestion. Nucleosome occupancy was assessed immediately prior to (one hour mock treatment) or after (one hour steroid treatment) LFY nuclear entry. All MNase-seq datasets were of high quality on the basis of characteristic phased nucleosome occupancy (Supplementary Fig. 4a-c).

To identify LFY binding events at nucleosome occupied sites, we called significant nucleosomes using DANPOS (occupancy qval ≤10$^{-50}$) in the MNase-seq datasets. We next compared LFY binding and nucleosome occupancy in a 2 kb region centered on the significant LFY peaks and ranked on the signal strength of the

nucleosomes present under the peak summit region (Fig. 2b, c, Supplementary Fig. 4d). Immediately prior to LFY nuclear entry, most LFY binding sites were nucleosome occupied (68%, Fig. 2b). Similar results were obtained with high MNase digest or when we probed nucleosome occupancy immediately after LFY binding (Fig. 2c, Supplementary Fig. 4d, Supplementary Data 1). The extent of the overlap between LFY binding and nucleosome occupancy (Fig. 2b, c, Supplementary Fig. 4d) is similar to that described for Sox2[14] and FoxA2[16], supporting the notion that LFY can bind to nucleosomal DNA in vivo.

Genes at which the LFY binding site was nucleosome occupied include *AP1*, as expected (Fig. 2d, Supplementary Fig. 4e). At another known LFY target, *REGULATOR OF AXILLARY MERISTEMS 1* (*RAX1*)[62], LFY bound naked DNA (Fig. 2d, Supplementary Fig. 4e). RAX1 promotes meristem growth prior

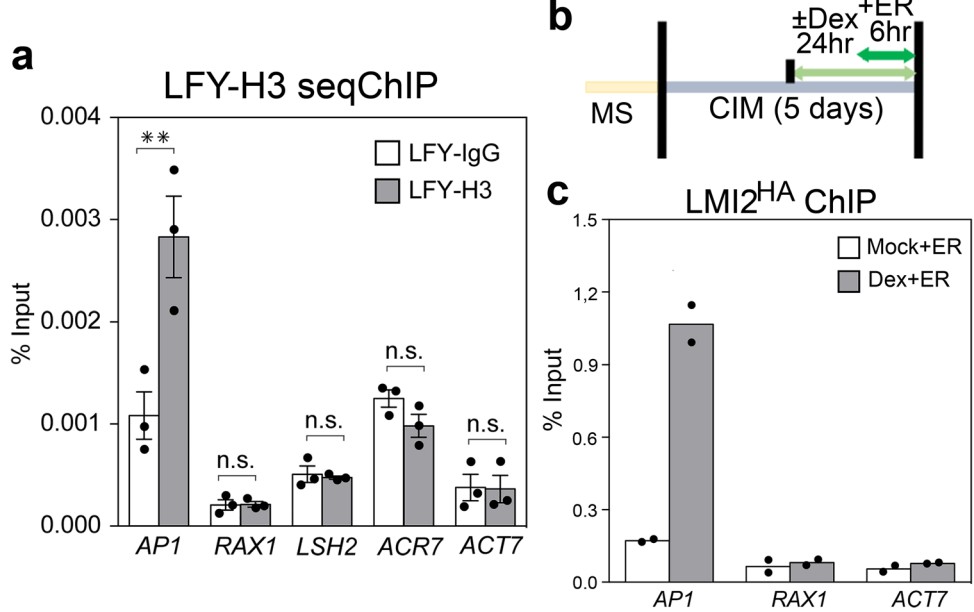

**Fig. 3 LFY binds nucleosomal DNA at the *AP1* locus and facilitates LMI2 chromatin access. a** Sequential LFY and histone H3 or IgG control ChIP to test LFY binding to nucleosomal DNA. Root explants were treated for 24 h with dexamethasone (Dex) solution. *p* value ** =0.009 (*AP1*), not significant (n.s.); p = 0.47 (*RAX1*), p = 0.36 (*LSH2*) p = 0.94 (*ACR7*), p = 0.77 (*ACT7*), one-tailed students' t-test, n = 3 independent biological experiments. Mean ± SEM. **b** Experimental setup to test LMI2-HA^ER recruitment to the *AP1* locus in root explants. 35S:LFY-GR root explants were either treated with (Dex) or without (Mock) steroid prior to estradiol (ER) induction of LMI2-HA^ER. **c** LMI2-HA^ER ChIP-qPCR at LFY bound sites. Black dots depict means from n = 2 independent biological experiments.

to onset of flower formation[62]. The housekeeping gene *ACTIN7* was neither bound by LFY nor had nucleosomes positioned over regulatory regions and was chosen as a negative control locus (Fig. 2d, Supplementary Fig. 4e).

To further test the ability of LFY to associate with nucleosome occupied regions in vivo, we conducted sequential ChIP in root explants using chromatin sonicated to nucleosome sized, 150 base-pair, DNA fragments[63] (Supplementary Fig. 5a). After initial immunoprecipitation for LFY, we dissociated the antigen/chromatin complex from the antibodies and subjected the chromatin to a second round of ChIP using a commercial anti H3 histone antibody. Sequential ChIP uncovered significant enrichment of H3 at *AP1*, but not at *RAX1* or the *ACT7* control locus (Fig. 3a). We probed additional loci where the LFY bound site was not (*LSH2*) or partially (*ACR7*) nucleosome occupied on the basis of our ChIP-seq and MNase-seq analyses (Supplementary Fig. 4e). Neither *LSH2* nor *ACR7* displayed H3 enrichment by sequential ChIP (Fig. 3a). The combined data confirm that LFY binds to its target site at the *AP1* locus in the context of a nucleosome in vivo.

Because the LMI2 binding motif is very close to that of LFY at the *AP1* regulatory region, (Fig. 1a), this provides us with the opportunity to examine whether LMI2 can associate with its target sequence at the *AP1* locus in the context of a nucleosome in vivo. After introducing an estradiol inducible version of LMI2 (LMI2-3HA^ER) into the 35S::LFY-GR genetic background, we treated root explants either with mock solution (no LFY chromatin association) or with dexamethasone (LFY chromatin association)[34], before inducing *LMI2-3HA^ER* expression by estradiol treatment (Fig. 3b, Supplementary Fig. 3b). Anti-HA ChIP-qPCR revealed that LMI2 bound to the *AP1* locus only after LFY activation (Fig. 3c). Thus without LFY, LMI2 does not associate with its binding site in the context of a nucleosome in vivo.

Next, we examined whether LFY binds nucleosome occupied sites in tissues where it triggers the switch from branch to flower fate[31]. Using public LFY ChIP-seq and MNase-seq datasets from stages when flowers form[46,64], we found that as in root explants, the majority (60%) of the LFY binding peaks overlapped with a nucleosome (Fig. 4a, b). By contrast, analysis of inflorescence ChIP-seq data for the B3 domain transcription factor AUXIN RESPONSE FACTOR3/ETTIN (ARF3/ETT)[64,65] uncovered that ETT preferentially binds naked DNA (9% of the binding peaks overlapped with a nucleosome, Supplementary Fig. 6a). Finally, we tested whether LMI2 can bind to the *AP1* locus in inflorescences, the physiological context for its activity. After mock or dexamethasone treatment of 25-day-old 35S::LFY-GR inflorescences grown in florally noninductive photoperiod[66], we induced LMI2-HA^ER. As in root explants, LMI2 bound to the *AP1* locus chromatin only after LFY activation (Fig. 4c). Thus, among the transcription factors examined, LFY alone strongly associates with target motifs occupied by nucleosomes in inflorescences.

**LFY binding to chromatin enriched DNA promotes floral fate in root explants**. Pioneer transcription factors enable gene expression changes in the context of closed chromatin by allowing binding of additional non-pioneer transcription factors and by directly or indirectly opening the chromatin at target loci[18,21,67,68]. Analysis of histone modifications in root explants[69] revealed that in the absence of LFY, both the *AP1* and the *RAX1* loci are marked by the repressive histone modification H3K27me3, while no H3K27me3 was present at the *ACT7* locus (Fig. 5a). Conversely, *ACT7* was significantly associated with the active H3K4me3 histone modification, which was absent from the *AP1* and *RAX1* loci (Fig. 5a). To monitor gene expression changes triggered by LFY, we next conducted a time-course RNA-seq analysis 1, 6, or 24 h after dexamethasone or mock treatment in root explants (Supplementary Fig. 7). Transcriptomes of dexamethasone and mock treated root explants began to differ from each other 24 h after LFY activation

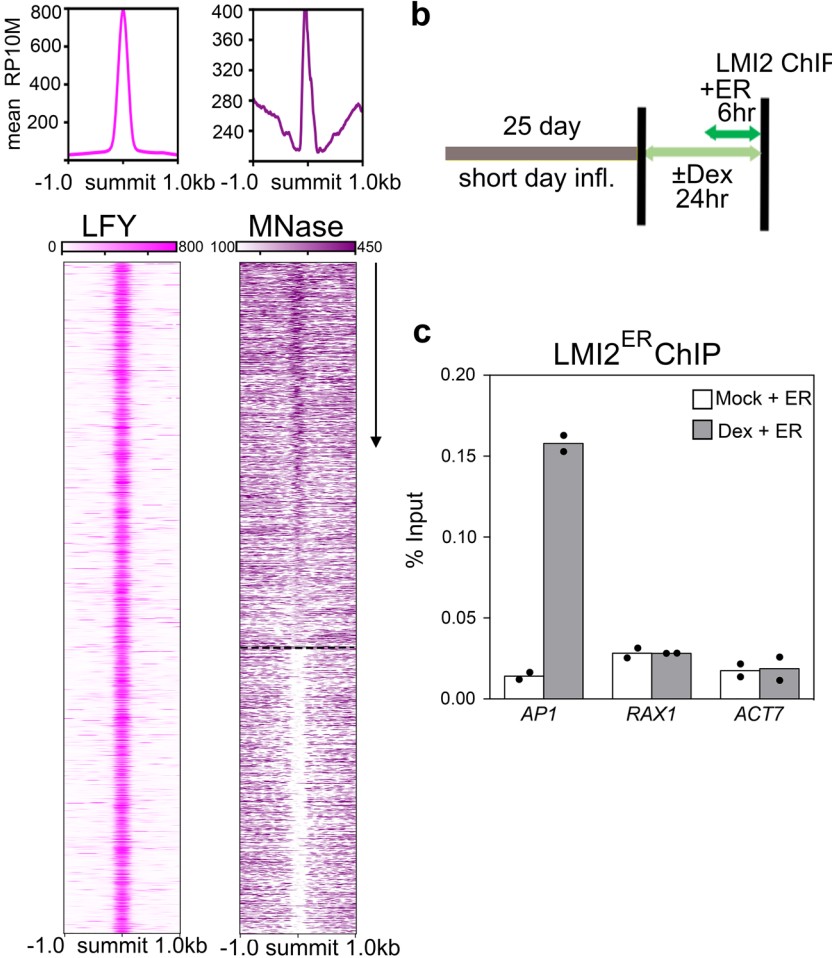

**Fig. 4 LFY binds to nucleosomes during flower formation. a** Top: mean signals of significant LFY ChIP-seq (RP10M; MACS2 $q$ value < $10^{-465}$)[46] or nucleosome occupancy (MNase-seq; DANPOS $q$ value < $10^{-300}$)[64] in the region ±1 kb of significant LFY peak summits. Bottom: Heatmaps for LFY ChIP-seq[46] and MNase-seq[64] in a 2 kb region centered on significant LFY peak summits and ranked by nucleosome occupancy levels (arrow) in the LFY peak summit region (±75 bp). Dotted line separates nucleosome occupied (top) from nucleosome free (bottom) LFY binding sites. **b** Experimental setup to test LMI2-HA^ER recruitment to the *AP1* locus in inflorescences. 35 S:LFY-GR inflorescences grown in non-inductive photoperiod (short day) were treated with steroid (Dex) or mock solution prior to estradiol (ER) induction of LMI2-HA^ER. **c** LMI2-HA^ER ChIP-qPCR at *AP1*, *RAX1* and *ACT7* loci. Black dots depict means from $n = 2$ independent biological experiments. The % input for LMI2 ChIP is lower than in root explants (Fig. 2c) as fewer cells are responsive to reprogramming in the inflorescence[34]. See also Supplementary Fig. 6.

(Supplementary Fig. 7a). *RAX1* was upregulated significantly 6 h (weakly) or 24 h (strongly) after steroid application, while *AP1* was significantly and strongly induced only 24 h after dexamethasone application (Fig. 5b, Supplementary Fig. 7b). In total we identified 5, 33, and 302 LFY bound and differentially expressed genes 1, 6, or 24 h after LFY activation, respectively (DESeq2 q-value ≤ 0.01 Supplementary Fig. 7c). The direct LFY regulated targets in root explants overlapped significantly with known direct LFY targets (Supplementary Fig. 7c)[57]. We next divided these direct LFY regulated targets into those where the LFY binding site was nucleosome occupied and those where the LFY binding site was nucleosome free. The two groups of genes showed similar expression fold changes in response to dexamethasone treatment at all timepoints assayed (Supplementary Fig. 7d). Despite their similar behavior, on the basis of gene ontology term enrichment analysis (AgriGO2 GOslim) only LFY regulated targets where LFY bound to a nucleosome occupied binding sites were linked to flower development (Fig. 5c, Supplementary data 2). Thus, LFY binding in the context of nucleosome-enriched chromatin activates flower related genes in root explants.

**LFY triggers local chromatin opening**. The helix-turn-helix DNA binding domain of LFY has structural similarity with linker histone H1 (Fig. 6a) and like LFY, the linker histone contacts the nucleosome near the dyad (Fig. 1a)[3]. Linker histones compact chromatin and H1 loss triggers precocious flowering in Arabidopsis[3,70]. To probe for chromatin opening by LFY, we therefore next determined occupancy of the H1 linker histone in the absence and presence of LFY. Anti H1 ChIP-qPCR 24 h after LFY activation compared to mock treatment revealed a strong reduction of linker histone occupancy at the LFY binding site of the *AP1* locus, but not at *RAX1* or *ACT7* (Fig. 6b, Supplementary Fig. 5b). Thus, LFY binding leads to loss of H1 linker histone at *AP1*. In developing flowers, LFY directly recruits the BRAHMA and the SPLAYED SWI/SNF chromatin remodeling ATPases to overcome Polycomb repression for flower patterning[71]. It is not known whether LFY recruits SWI/SNF complexes to activate floral fate. After introducing a tagged version of SWI3B—a core component of both the BRAHMA and the SPLAYED SWI/SNF complexes[72]—into LFY-GR plants, we examined SWI3B occupancy 24 h after mock or dexamethasone treatment. LFY activation lead to significant SWI3B recruitment to the *AP1* locus,

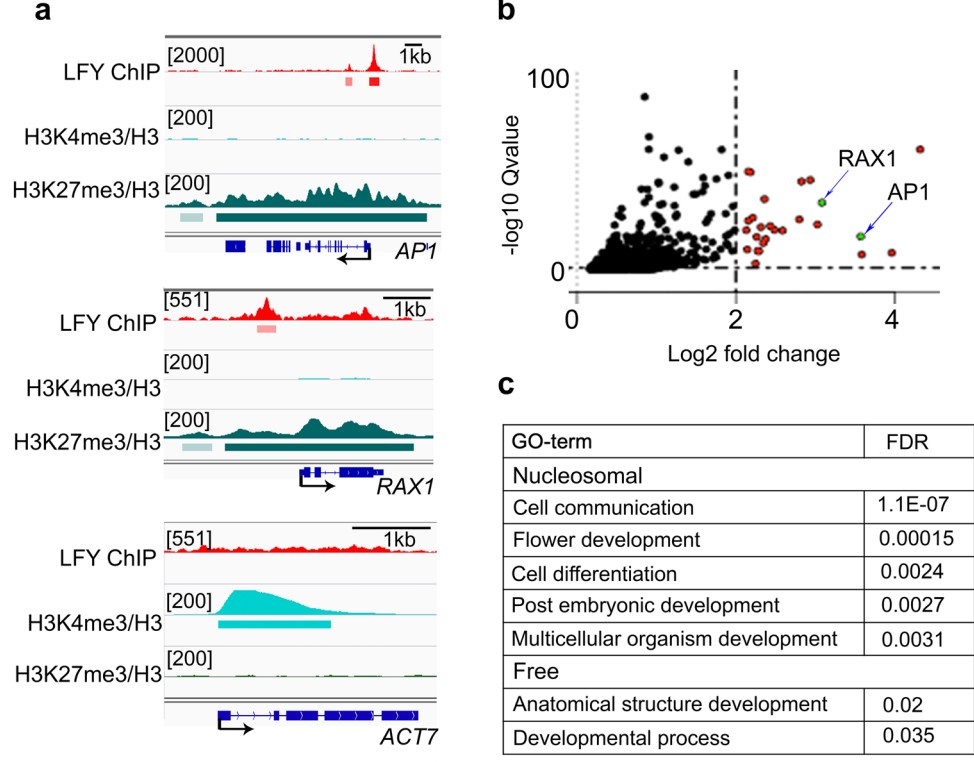

**Fig. 5 LFY binding in the context of nucleosome-enriched chromatin activates flower related genes. a** Browser view of LFY occupancy (this study) and presence of H3K27me3/H3 or H3K4me3/H3 based on analysis of a public ChIPseq dataset[69] in root explants at *AP1*, *RAX1*, and *ACT7*. Significant ChIP peaks (*q* value ≤ $10^{-10}$ MACS2 for LFY and *q* value ≤ $10^{-100}$ for histone modifications) are marked by horizontal bars, with the color saturation proportional to the negative log 10 *q* value (as for the narrowPeak file format in ENCODE). **b** Volcano plot of genes differentially expressed 24 h after dexamethasone relative to mock treatment (*n* = 2042, adjusted DESeq2 *p* value ≤ 0.01). Red color denotes log 2 FC ≥ 2, arrows point to *AP1* and *RAX1*. **c** Significantly enriched development related Gene Ontology terms for LFY regulated genes whose LFY binding site was nucleosome occupied (Nucleosomal *n* = 202, FDR < 0.005; top) or nucleosome free (Free *n* = 101, FDR < 0.05; bottom). All significant GO terms identified are listed in Supplementary Data 2. See also Supplementary Fig. 7.

but not to *RAX1* or *ACT7* (Fig. 6c, Supplementary Fig. 5c). Thus, LFY initiates local chromatin changes upon associating with its nucleosome bound target sites at the *AP1* locus. Finally, we tested for broad changes in chromatin accessibility at known DNase hypersensitive sites[73]. We conducted formaldehyde assisted identification of regulatory elements (FAIRE)[74] followed by quantitative PCR at the *AP1*, *RAX1* and *ACT7* loci. We did not observe increased accessibility at any of the loci tested 24 h after LFY activation (Fig. 6d, Supplementary Fig. 5d). However, 5 days after LFY activation, we observed a significant increase in chromatin accessibility at the *AP1* locus, but not at *RAX1* or *ACT7* (Fig. 6e, Supplementary Fig. 5e). The delayed chromatin opening is consistent with the continued increase in *AP1* expression until day five after LFY upregulation (Supplementary Fig. 2d).

**LFY DNA contact helix and nucleosomal binding sites are characteristic of pioneer transcription factors.** Studies of animal pioneer transcription factors have highlighted structural properties of the DNA recognition moieties of transcription factors critical for pioneer activity. In particular, pioneer transcription factors have short DNA recognition helices, which make contacts on one face of the DNA[14,15]. This leaves the majority of the circumference of the DNA solvent exposed, consistent with simultaneous transcription factor and histone octamer occupancy[14,15]. LFY binds DNA as a monomer and as a homo-dimer[57]. Analysis of the structure of the LFY bound to DNA[54] revealed that, both as a monomer and a dimer, LFY makes very shallow contacts on one face of the DNA (Fig. 7a), leaving more

than 50% of the DNA surface free to interact with histones in a nucleosome. Indeed, such short anchoring alpha helices are frequently found in strong nucleosome binders like FoxA[15]. Since ETT/ARF3 preferentially bound naked DNA (Supplementary Fig. 6a), we wished to examine the structure of its DNA contact domain bound to DNA. Structural data is available only for DNA contact domains of AUXIN RESPONSIVE FACTOR1 (ARF1)[75], which is closely related to ARF3/ETT[76]. The ARF1 monomer or homodimer DNA binding domain is comprised of beta sheets and disordered loops that interact with one face of the DNA (Supplementary Fig. 6b). Compared to LFY, the ARF1 DNA contact domains extend further into the DNA, especially in the ARF1 dimer (Supplementary Fig. 6b). Whether these ARF1 contacts would preclude simultaneous histone interactions is not clear. We conclude that LFY associates with cognate motifs in the context of nucleosomes and that this may be enabled by structural properties of its DNA contact helix.

Most pioneer transcription factors bind target sites in the context of nucleosomes as well in free DNA in vivo[14–17]. However, the types of binding motifs they associate with in each case often differ. Binding sites in free DNA tend to be closer in sequence to the consensus motif, while sites bound in nucleosomes generally deviate more from the consensus[14,17]. LFY binds a palindromic sequence, which at its core has the sequence CCANTGG[54,56,57]. As described for Oct4 and Sox2[14], the top LFY motif identified by de novo motif analysis from the naked DNA more closely resembles the consensus than does the top motif identified from the nucleosome occupied binding sites

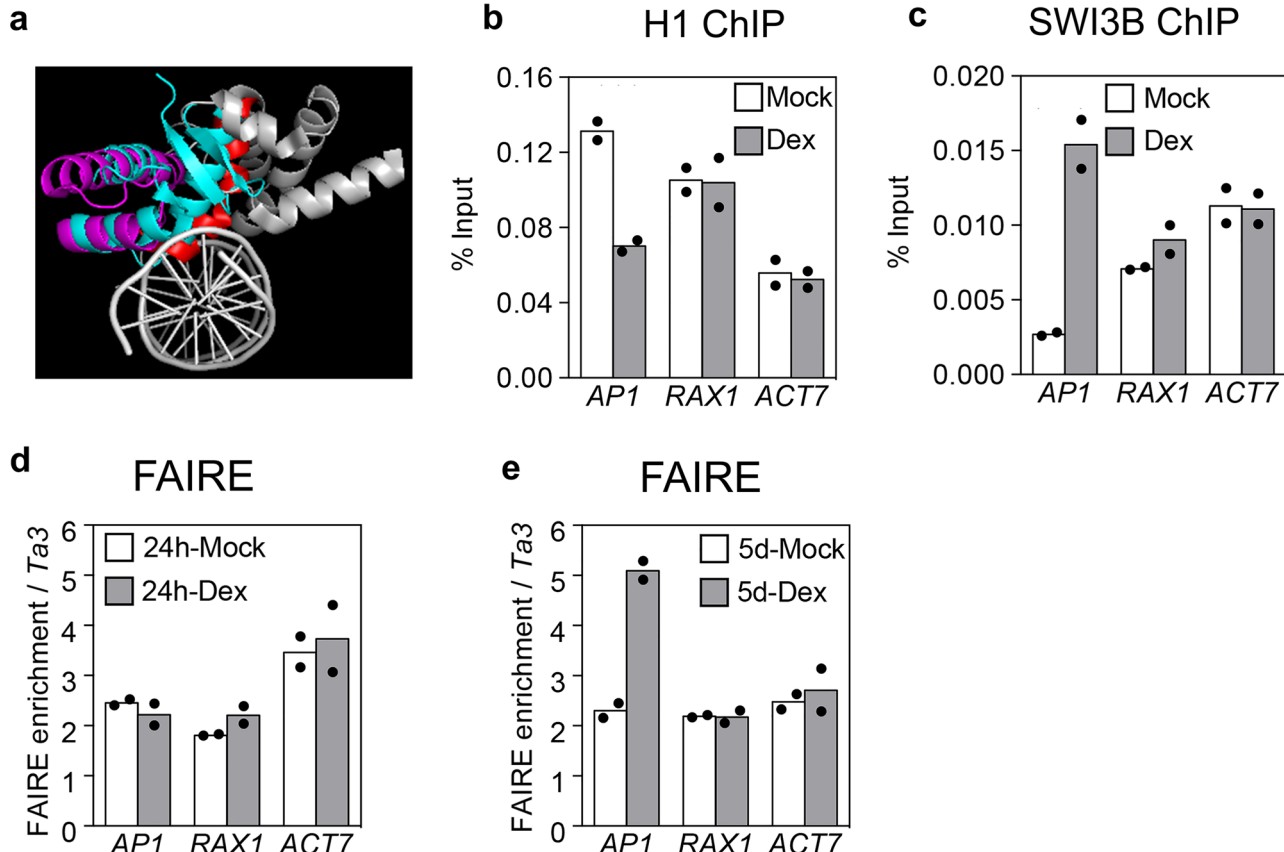

**Fig. 6 LFY displaces histone H1 and recruits chromatin remodelers. a** Comparison of the structure of linker histone H1 (PDB: 5NL0) and of the LFY DNA binding domain (PDB: 2VY1) bound to DNA. Red: All base contacting residues, including the LFY anchoring helix. Warm pink and red: helix-turn-helix DNA binding domain of LFY. Gray: remainder of the LFY C-terminal domain[54]. Turquoise: H1 linker histone. **b** Histone H1 ChIP-qPCR at the LFY bound sites of *AP1*, *RAX1* and *ACT7* 24 h after dexamethasone (Dex) or control (Mock) treatment. Black dots: means from n = 2 independent biological experiments. **c** ChIP-qPCR of the SYD and BRM SWI/SNF complex subunit SWI3B at the LFY bound sites of *AP1*, *RAX1* and *ACT7* 24 h after dexamethasone (Dex) or control (Mock) treatment. **d, e** FAIRE qPCR at known DNase hypersensitive sites[73] of the loci indicated 24 h after LFY activation (**d**) or 5-days after LFY activation (**e**) relative to the control (Mock treated plants). Black dots: means from n = 2 independent biological experiments. See also Supplementary Fig. 5.

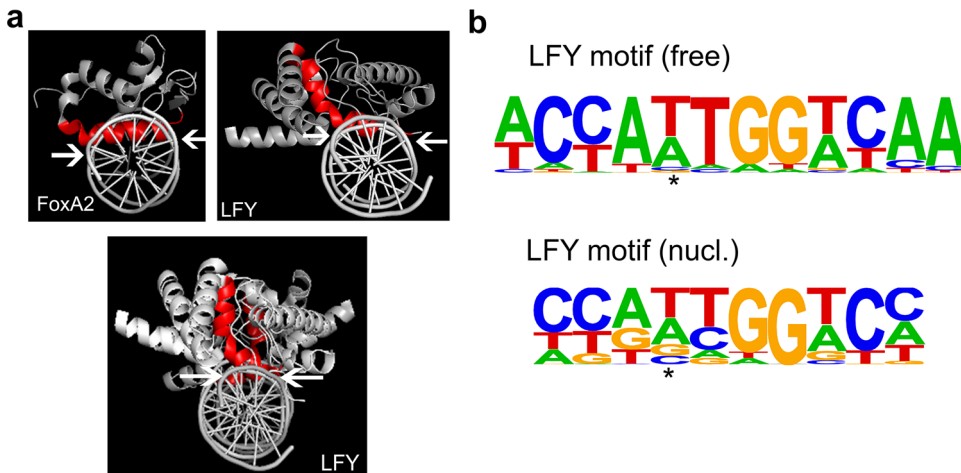

**Fig. 7 LFY DNA contact helix and motif preference. a** Structure of the known pioneer transcription factor FoxA (PDB:5X07 [https://www.rcsb.org/structure/5X07]), and LFY monomer (top right) or dimer (bottom) (PDB: 2VY1) bound to DNA. Red: All base contacting residues, including the DNA anchoring helix. Arrows delineate the DNA region contacted. **b** Position weight matrices of top motifs identified by de novo motif analysis (HOMER) in naked DNA (LFY motif (free); *p* value =1E-43) or in nucleosomes (LFY motif (nucl.); *p* value =1E-48). The nucleosomal LFY motif diverges more from the known CCANTGG[56, 57] core consensus motif. Asterisk: Center of the palindrome. See also Supplementary Fig. 6.

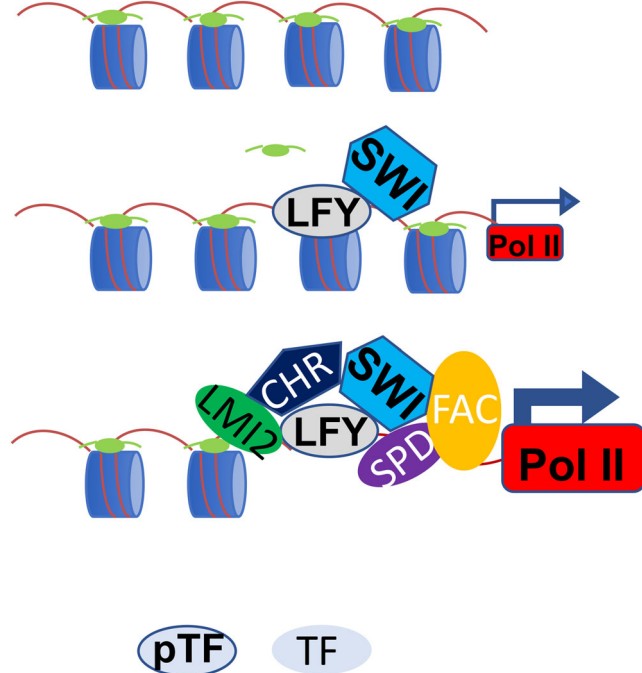

**Fig. 8 Hierarchical model for root explant reprogramming to floral fate by the LFY pioneer transcription factor.** LFY binds to a nucleosome occupied target site and locally opens chromatin by displacing H1 linker histones (light green ellipse) and by recruiting SWI/SNF chromatin remodelers (SWI hexagon). This leads to low initial activation of gene expression (small arrow and RNA polymerase II (Pol II)). The local chromatin unlocking enables additional transcription factors to bind the locus, these include LMI2, SPL9/DELLA (SPD) and the florigen activation complex (FAC). See text for details. This in turn leads to recruitment of other chromatin regulators (CHR pentagon), increased gene activation (large arrow and Pol II), accumulation of the floral fate commitment factor AP1 and cell fate reprogramming. pTF pioneer transcription factor. TF non-pioneer transcription factor.

(Fig. 7b). It is thought that the degenerate motifs require shorter DNA contact helices and thus facilitate simultaneous transcription factor and histone contacts[14,15]. This is further aided by the nucleosomal binding motifs themselves being shorter[14], as we find here for LFY (Fig. 7b).

## Discussion

Here we identify LFY as a pioneer transcription factor. In vitro, LFY binds with high affinity to its cognate binding motif in a native DNA fragment from the *AP1* locus if we assemble this DNA into a nucleosome. LFY also binds nucleosome occupied DNA in vivo. Evidence in support of this conclusion comes from combined ChIP-seq and MNase-seq analyses in root explants and inflorescences, as well as from sequential LFY and histone H3 ChIPseq in root explants. At its key target locus *AP1*, LFY displaces linker histone H1 and recruits SWI/SNF chromatin remodelers to initiate *AP1* expression, but larger-scale locus opening occurs later, concomitant with increased *AP1* upregulation. Both recent studies in animals[14,17,24,77,78], and our findings suggest a hierarchical model for pioneer factors which 'license' transcriptional reprogramming but may not - by themselves - trigger strong chromatin opening (Fig. 8). Instead they 'unlock' the chromatin to initiate a series of events, including binding of additional, non-pioneer, transcription factors and further recruitment of chromatin regulators, that collectively result in enhanced chromatin accessibility, strong transcriptional

activation and altered cell fate (Fig. 8). Pioneer activity may be especially important for target genes with restricted spatio-temporal expression that commit cells to new fates.

LFY is a master regulator of onset of flower formation and can reduce time to formation of the first flower in trees from decades to months[32]. Although LFY alone is sufficient to activate *AP1*[79,80], accumulation of this floral commitment factor is delayed relative LFY activity. Under floral inductive conditions in inflorescences, *AP1* upregulation occurs 2–3 days after that of *LFY*[52,81,82]. The delay in *AP1* upregulation is of biological significance as it enables formation of branches prior to the irreversible switch to flower fate[52]. The duration of the delay in flower formation tunes the inflorescence architecture to environmental cues to enhance reproductive success[83]. Molecularly, the delayed *AP1* upregulation is attributable—at least in part—to a requirement for co-factors activated by LFY in 'and' logic feed-forward loops (FFLs). Such FFLs not only make biological processes more robust to noisy stimuli such as environmental cues, but they also represent temporal delay elements[84]. In one FFL, LFY activates LMI2, which upregulates *AP1* together with LFY[51]. In another FFL, LFY directly triggers reduced accumulation of bioactive gibberellin hormone; this stabilizes the DELLA/ SQUAMOSA BINDING PROTEIN9 (SPL9) transcriptional complex, which activates *AP1* in parallel with LFY[52].

Combining these and additional prior findings with those of our current study, we propose that the pioneer transcription factor *LFY* establishes competency for cell fate reprogramming to floral fate by associating with the nucleosome occupied binding site at the *AP1* locus where it opens chromatin locally, concomitant with initial (low level) *AP1* upregulation (Fig. 8, Supplementary Fig. 2d). This local chromatin opening may be the result of direct changes in DNA-histone contacts caused by LFY, as described for other pioneer transcription factors[85]. Alternatively, local chromatin opening may be caused by changes in nucleosome-linker or nucleosome–nucleosome interaction due to linker histone H1 displacement by LFY (Fig. 5c) or by SWI/SNF mediated chromatin remodelling (Fig. 5d). That LFY 'unlocks' the chromatin at the *AP1* locus in this manner is supported by the critical role of the LFY binding site for *AP1* locus activation[53]. As a result, LFY pioneer activity enables binding of additional (likely non-pioneer) transcription factors; these include not only LMI2, but also SPL9/DELLA and the florigen activator complex, which all bind near the LFY binding site at the *AP1* locus to activate gene expression[51,52,60,81,86–89]. We propose that the activity of these LFY co-factors, in a manner not yet understood, contributes to further opening of the *AP1* locus chromatin, enables strong *AP1* upregulation and reprogramming to floral fate (Fig. 8). In summary, the delay between LFY pioneer factor binding and local chromatin opening at *AP1* on one hand, and broad locus accessibility and strong *AP1* upregulation on the other, is likely attributable to the timed accumulation (FFLs) and hierarchical recruitment of the LFY co-factors LMI2, SPL9/DELLA and the florigen activation complex to the *AP1* locus. This ensures correct timing of the irreversible switch to flower formation, which is critical for reproductive success.

Animal pioneer transcription factors play important roles in developmental reprogramming in vivo, generation of iPS cells and trans-differentiation, enabled by their unique ability to bind cis motifs even when buried deep in the nucleosome (near the dyad)[15,16,18,20,21,80]. While first 'rules' or characteristics of pioneering activity of transcription factors are emerging[11,14,15,17,77], it is far from clear what their unique or defining set of properties is. For example, some pioneering factors act cooperatively with other transcription factors, but it is not known how widespread such interactions are[17,24,90]. The LFY pioneer transcription factor shares many properties with animal pioneer factors in addition to

the ability to bind nucleosomes in vitro and in vivo. Like the pioneer transcription factors Oct4, Ebf1, and Rap1[78,91,92], LFY recruits SWI/SNF chromatin remodeling complexes to key target loci. Like FoxA[5,16], LFY displaces linker histone H1 in vivo. Like the pioneer transcription factors Pax7[24,93], PHA-4[77], and Rap1[78], LFY rapidly associates with nucleosome-occupied binding sites at target loci, but chromatin opening is delayed relative to binding. Finally, the LFY DNA contact helix shares structural properties with strong nucleosome binders[15] and LFY binds weaker motifs in nucleosomes, as described for Oct4 and Sox2[14]. Unique properties of the LFY pioneer transcription factor include its ability to bind both fragile and stable nucleosomes (this study) and to contact a consensus motif half site as a monomer in vitro and in vivo[54,57]. Combined with the fact that the LFY binding site is palindromic[54,56,57], this enables LFY to bind its cognate motif in a nucleosome even if the DNA is rotated 180° to face the histone octamers (altered rotational nucleosome positioning)[94,95]. Moreover, LFY bound sites cluster around the transcription start site[57], which enhances nucleosome positioning[96,97]. These combined characteristics make the pioneer transcription factor LFY extremely well suited to license key developmental transitions.

Plants evolved multicellularity independently from animals and can change their final body plan in response to environmental cues[98], suggesting that pioneer transcription factors may be more prevalent in this kingdom. Our study sets the stage for identification and validation of further plant pioneer transcription factors, characterization of which may not only help understand the basis for plant developmental plasticity but should also contribute to elucidation of the rules for pioneer activity by transcription factors in multicellular eukaryotes and enable their future engineering for enhanced cell fate reprogramming.

## Methods

**Plant materials.** 35S::LFY-GR is in the *Arabidopsis thaliana* wild-type accession Landsberg *erecta*[48]. LMI2[ER] in binary vector pMDC7[99] and SWI3B-3xHA in binary vector pGWB1[100] were transformed into 35 S::LFY-GR plants. For experiments in inflorescence, plants were grown in soil at 22 °C short-day photoperiod (SD, 8 h light/16 h dark, 120 μmol/m²s). For all other experiments, plants were grown on 1/2 MS plates (half strength Murashige and Skoog medium supplemented with 0.5 g/L of MES monohydrate, pH = 5.7, 0.8% phytoagar) at 22 °C in long-day photoperiod (LD, 16 h light/8 h dark, 100 μmol/m²s). *Arabidopsis* seeds were stratified at 4 °C for 3 days.

To obtain root explants, roots from 3-week-old seedlings were harvested and placed on callus inducing medium (CIM) (3.08 g/L Gamborgs B5 salts, 20 g/L glucose, 0.1 g/L myo-inositol, 0.5 g/L MES, pH = 5.7, supplemented with B5 vitamin solution, 0.5 mg/L 2,4D and 0.05 mg/L kinetin)[34] for 5 days. Dexamethasone (5 μM final concentration in 0.1% ethanol; Sigma Aldrich, D4902) or mock (0.1% ethanol) treatments lasted the entire duration (5 day) or commenced 24 h, 6 h, or 1 h before the end of the 5th day. Tissue was harvested at ZT 6.

**Constructs.** *LMI2* fused with 3 times Hemagglutinin (HA) was amplified from a published construct[51] and cloned into *pENTRD-TOPO* (Invitrogen, K243520). *pENTR LMI2-3HA* was moved to *pMDC7*[99] by LR reaction (Invitrogen, 11791-020). For *pSWI3B::SWI3B-HA*, a genomic fragment covering 311 bp upstream of the transcription start site and genomic SW3B coding region (2124 bp) excluding the stop codon were fused to 3xHA plus stop codon and the 163 bp 3′UTR sequence (primers listed in Supplementary Table 1). The fragments were inserted into *pENTR-3C* (Invitrogen, A10464) and the resulting pSWI3B::SWI3B-3xHA-3′ UTR construct was cloned into the binary vector *pGWB1* by LR reaction (Invitrogen, 11791-020). Constructs were introduced into *Agrobacterium* strain GV3101 for plant transformation[101–103].

For protein expression, the coding sequences of LFY and LMI2 were moved into pE-SUMOpro[104,105] using Gibson assembly (NEB, E2611) with primers listed in Supplementary Table 1. The constructs were introduced into *E.coli* Rosetta™2 (DE3) for protein expression.

**ChIPqPCR.** For LMI2 ChIP-qPCR, root explants on CIM were treated with 5 μM dexamethasone or mock solution for 18 h followed by addition of beta-estradiol (5 μM in 0.1% ethanol Sigma Aldrich, E2758) and a further incubation for 6 h. 100 seedling plates yield about 1 g of tissue. For each sample, 2 grams of tissue was harvested at the end of day 5 on CIM. For LMI2 ChIP-qPCR in inflorescence, for

each sample, 2 grams of shoot apices from 25-day-old, short-day grown LMI2[ER]35S: LFY-GR plants were harvested from 4 trays of plants. After trimming, shoot apices were submerged in 5 μM dexamethasone or mock solution for 18 h followed by addition of 5 μM beta-estradiol and a further incubation for 6 h. For LFY ChIP-qPCR and ChIP-seq, 35 S::LFY-GR root explants were treated with 5 μM dexamethasone for 1 hour prior to the end of day 5 on CIM. For each sample, 1.2 g of root explant tissue was harvested. For H1 and SWI3B-3xHA ChIP-qPCR, root explants were treated with 5 μM dexamethasone or mock solution for 24 h. For each sample, 0.6 and 1.2 grams of tissue was harvest for H1 and SWI3B ChIP-seq, respectively. At least two biological replicates were generated for each experimental condition.

ChIP was conducted using our protocol[106] and the following antibodies: rabbit polyclonal anti-LFY antibodies[48,57], mouse anti-HA antibody (ROCHE, 12CA5), Mouse IgG2bκ Anti-HA antibody (Abcam, 46540), rabbit anti-histone H1 antibody (Abcam, 61177), rabbit anti-histone H3 antibody (Abcam, 18521), and rabbit anti mouse IgG (Abcam, 46540). ChIP-qPCR was performed using Platinum Taq DNA Polymerase (Invitrogen, 10966034) and EvaGreen dye (Biotium, 31000). Inputs from each sample were used to generate the standard curve to compute sample enrichments[106]. Primer sequences for ChIP-qPCR are listed in Supplementary Table 1.

For sequential-ChIP, four ChIP reactions from 0.5 g root explant tissue each were combined into one biological replicate. Three such biological replicates were analyzed after 24 h steroid application (Fig. 3a). A total of 15 cycles of sonication were used to obtain chromatin fragments of ~150 bp to probe LFY and histone co-occupancy. The first (LFY) overnight immunoprecipitation and the subsequent wash steps were performed as in a published protocol[106]. The eluted chromatin was next immunoprecipitated for histone H3 or IgG following an established protocol[63] using commercial antibodies (above). The anti-H3 or IgG chromatin bound to protein G beads was eluted with 50 mM Tris–HCl pH 8.0, 5 mM EDTA, 20 mM DTT, and 1% SDS at 37 °C for 30 min.

**ChIP-seq and data analysis.** For LFY ChIP-seq, each biological replicate was generated by pooling four individual ChIP reactions (each consisting of 0.6 g of root explant tissue) prior to DNA purification on MinElute PCR columns (Qiagen, 28004). Three biological replicates were generated for each condition. Dual index libraries were prepared for the six ChIP samples (three mock and three treatment) and three input samples using the SMARTer ThruPLEX DNA-Seq Kit (Takara Bio, R400406). After quantifying libraries using NEBNext Library Quant Kit for Illumina (NEB, E7630), libraries were pooled based on desired read depth. Single-end sequencing was conducted using NextSeq 550/500 High Output Kit v2 (Illumina, TG-160-2005) on the NextSeq500 platform (Illumina).

After trimming adapter sequences and low-quality reads by Trimmomatic (v0.32)[107], FastQC (v0.11.5) was performed on trimmed reads to confirm high-quality sequences[108]. Sequencing reads were then mapped to release 10 of the Arabidopsis Genome (TAIR10)[109] by Bowtie2 (—phred33 -q -x) v2.3.1[110]. Next, uniquely mapped reads (quality score MAPQ ≤10−30 (Samtools v1.7))[111] were processed following ENCODE guidelines. Significant ChIP peaks and summits (summit q-value ≤10−10) were identified in the pooled dexamethasone-treated samples using the pooled mock-treated samples as controls in MACS2 (v2.1.2)[112,113] (—keep-dup auto —nomodel —extsize 138 —call-summits -g 101274395). This yielded 1177 significant LFY peaks. For quality control, Spearman correlation coefficients of the reads in the LFY peak regions of each biological replicate were compared using deepTools (v3.1.2)[114,115]. De novo motif analysis was conducted using HOMER v4.1[116] for MACS2 q-value ≤ 10−10 peak summits (±150 base pairs) compared to genome matched background (unbound regions from similar genomic locations as the peak summits), for an example of this approach see Refs. [57,117].

LFY ChIP-seq data (15day-old 35S::LFY)[46] was retrieved from GEO dataset GSE64245). Two replicates were analyzed. Inflorescence ETT ChIP-seq data[65] was obtained from EBI-ENA database accession number PRJEB19862). Three biological replicates and one negative control file were analyzed. Trimming, FastQC, filtering (low quality reads) and mapping were done as described for LFY ChIP in root explants (above). For inflorescence ChIP seq, peaks were called using MACS2 (v2.1.2)[112,113] using the same parameters as for the ChIP-seq analysis in root explants. Since no control files were available, we used more stringent criteria for peak calling (MACS2 summit q-value ≤ 10−465). This identified 1952 significant LFY binding peaks. For ETTIN ChIP seq, peaks were called using (summit q-value≤10−10) as described for LFY ChIP in root explants. A total of 670 significant peaks were identified.

Histone modification ChIP-seq data from root explants cultivated for 14 days on CIM[69] were retrieved from DDBJ Sequence Read Archive (DRA) under the accession number DRA008014 for histone H3, H3K4me3 and H3K27me3 in the Columbia ecotype. Trimming, FastQC, filtering for low quality reads and mapping were as described for LFY ChIP-seq (above). For all histone modification ChIP-seq, peaks were called using MACS2[112,113] callpeak command -f BAM —call-summits —keep-dup auto —nomodel —extsize 138 -g 101274395, using the histone H3 file as the control file and the respective histone modification file as the treatment file. For H3K4me3 and H3K27me3 peaks, q-value ≤ 10−100 was used for significance calling and a total of 6032 and 1359 peaks were identified, respectively.

**FAIRE-qPCR.** For FAIRE-qPCR, 35S:LFY-GR root explants were treated with 5 μM dexamethasone or mock solution for five-days or for the last 24 h on CIM. For

Fig. 5 tissue was flash frozen in liquid nitrogen. For each biological FAIRE replicate 0.4 g of frozen tissue was ground and fixed with 1% final formaldehyde for 6 min[118]. Control samples consisted of 0.2 g tissue per replicate processed without fixation. For Supplementary Fig. 5, tissue was crosslinked (FAIRE) immediately after steroid treatment before freezing and 0.8–1.2 gram of root explants was used for each biological replicate. In both cases, chromatin was isolated as for ChIP[106] and FAIRE-qPCR was conducted as previously described[119]. Primer sequences for FAIRE-qPCR were designed to query published DNase I hypersensitivity sites near each candidate locus[73] and are listed in Supplementary Table 1.

**MNase-qPCR, MNase-seq and data analysis**. For MNase-qPCR and MNase-seq, 35S::LFY-GR root explants were treated with 5 μM dexamethasone or mock solution for the last hour of the 5-day CIM incubation. 0.6 g of tissue was harvested per replicate into liquid nitrogen. Nuclei and chromatin were isolated following standard protocols[106,120–122] with the following modifications. After washing isolated nuclei twice with HBB buffer (25 mM Tris-Cl, pH 7.6, 0.44 M Sucrose, 10 nM MgCl₂, 0.1% Triton-X, 10 mM beta-ME), isolated chromatin was digested with 0.5 unit/μl or 0.05 unit/μl (final concentration) of Micrococcal Nuclease (Takara, 2910 A) for 3 min at 37 °C to obtain high or low digestion of chromatin, respectively. Subsequent steps were performed as in[121]. Mono-nucleosomes were excised from 1.5% agarose gels and purified using the QIAquick gel extraction kit (Qiagen, 28115). Purified DNA was diluted 50 times for qPCR analyses.

Tiling primers (each primer pair covering 80 bp with 40 bp overlap between neighboring primer sets) were used for MNase-qPCR spanning the *AP1* regulatory region and are listed in Supplementary Table 1. The standard curve was generated relative to nucleosome occupancy at the Gypsy (AT4G07700) retrotransposon[122]. Tiling primers were designed based on our analysis of an Arabidopsis vegetative stage MNase-seq dataset[55] retrieved from DDJB Sequence Read Archive (DRA) accession number SRP045236 samples SRX669229).

For MNase-seq (low MNase digestion in mock or steroid treated LFY-GR and high digestion MNase digestion in mock or steroid treated LFY-GR) two independent replicate libraries were constructed as described above for ChIPseq, quantified using NEBNext Library Quant Kit for Illumina (NEB, E7630), pooled and paired-end sequenced using NextSeq 550/500 High Output Kit v2 (Illumina, TG-160-2005) on the NextSeq500 platform (Illumina). Quality control and filtering were identical to the ChIP-seq analysis. Mapping was performed using the paired-end mode of bowtie (v1.2.2)[110] with —no-unal -S —chunkmbs 200 —best -m 1 parameters. Using the mapped reads, DANPOS (v3.0.0)[123] was employed (-q 50 smooth_width 10 -H 1 -m 1 —mifrsz 40 -u .1) to identify nucleosome occupied regions using q <$10^{-50}$ as cutoff on the replicates, except for low MNase digestion, dexamethasone treated, samples where the replicates were pooled prior analysis. A published MNase-seq data from inflorescences[64] (EBI-ENA database study PRJEB15184, samples SAMEA4386620 rep1 and rep2) was analyzed as described above except DANPOS q <$10^{-300}$ was used to control for higher background. A total of 317,312 (Low MNase Mock treatment), 237,480 (Low MNase Dex treatment), 482, 659 (High MNase Mock treatment), 376,790 (High MNase Dex treatment) and 438712 (Inflorescence)[64] significant nucleosomes were called.

To validate MNase digestions, read enrichment heatmaps[16] were generated for regions spanning ±1000 bp of the transcription start sites (TSS) of protein-coding genes. To build the heatmap, read counts were normalized by sequencing depth and counted in 20 bp bins for each gene. Gene regions were sorted by the sum of their normalized read values and averaged in groups of 5. Horizontal Bartlett smoothing[124] was applied using an 11-bin window.

Heatmaps comparing transcription factor and nucleosome occupancy were centered on the ChIP-seq summits and ranked by the DANPOS summit occupancy value of the nucleosome that most overlapped with the ChIP summit (±75 bp). ChIP or MNase signals ± 1 kb of the summit were visualized using deepTools v3.1.2[114,115]. To test the reproducibility of the three MNase-seq datasets, Pearson correlation coefficients of the reads from pooled bigwig files in the LFY peak regions were used. Matrices and graphs were generated using deepTools[114,115].

To classify ChIP peaks into nucleosome bound vs. naked DNA bound sites, significant ChIP summits were overlapped with significant nucleosome regions using BEDTools v2.26.0 *intersect* function[125] and separated into two region files: LFY binding site nucleosome occupied or LFY binding site nucleosome free. Binding peaks whose summit overlapped with a significant nucleosome based on the DANPOS thresholds described above in at least three of the four MNase datasets for root explants were considered nucleosome occupied.

**Protein purification, nucleosome assembly and EMSA**. pE-SUMOpro-LFY and pE-SUMOpro-LMI2 in *E.coli* Rosetta™2 (DE3) (Novagen, 71397) were induced during exponential growth with 1 mM IPTG followed by incubation at 18 °C overnight in LB supplemented with 0.2% glucose. Cell pellets were lysed using 500 mM NaCl, 20 mM Tris-HCl pH 8, 5 mM imidazole, 5% glycerol supplemented with cOmplete™ EDTA-free protease inhibitor cocktail (Roche, 04693132001), 2 mM MgCl2 and 1ul/ml benzonase nuclease (Millipore, E1014). After sonication on ice, cell debris was removed by centrifugation (13,000 × *g* for 30 min at 4 °C). Proteins were affinity purified from the supernatant using Ni-NTA agarose (Invitrogen, R90115) and passed through a HiTrap Q HP anion exchange

chromatography column (GE healthcare, 17115301). Protein concentrations were determined by SDS-PAGE, using a BSA standard curve run on the same gel.

Recombinant human full-length histones H2A, H2B, H3, and H4, which share 80–95% amino acid identity with their Arabidopsis counterparts, were expressed and purified[126]. A 152 bp fragment region from the *AP1* regulatory region (TAIR10 Chr1:25986457 – 25986608) with the LFY binding site in the center and containing the LMI2 binding site served as wild type nucleosomal template. The LFY binding site mutated fragment was generated by changing GGAAGGACCAGTGGTCCGTA to GGCAGGAAAAGTAATCCGCA, while the LMI2 binding site mutated fragment replaced CCGTCAAT with GGAGACCG. After synthesis (GeneScript), *AP1* regulatory region DNA fragments containing BamHI sites were cloned into pUC19. At least 1 mg of the resulting plasmids was isolated using Plasmid Mega Kit (Qiagen, 12181), digested with BamHI, followed by fragment purification by gel electrophoresis and dialysis[15]. Subsequently, we Cy5 labeled the three DNA probes (wild type, LFY binding site mutant, LMI2 binding site mutant)[117]. We conducted nucleosome assembly with Cy5-labeled DNA fragments by mixing purified core histone dimers and DNA at 1:1 molar ratio in 2 M NaCl supplemented with 0.1 mg/mL BSA[14,15]. Cy-5 labeled DNA was then assembled around core histones by stepwise dialysis with decreasing concentration of salt and urea as previously described[14,15]. Initially, assembled nucleosomes were run on a native gel to see whether multiple nucleosome bands formed—an additional 2 hr 42 °C heat shift was performed if multiple nucleosome conformations were present as previously described[14,15]. Glycerol gradients of the dialyzed assembled nucleosomes were employed to separate free DNA from nucleosomes and the fractions collected were then dialyzed in 10 mM Tris–HCl pH = 8.0 for 1 h in 4 °C to remove glycerol as previously described[14,15]. Finalized dialyzed nucleosomes were concentrated using Amicon Ultra-0.5 ml (ultracel 10 K) (Millipore, UFC501024) at 16,500 × *g* for 10 min in 4 °C.

DNA and nucleosomal binding reactions were performed using established procedures[14,15]. Briefly, Cy-5 labeled DNA fragments and nucleosomes were diluted to 10 nM concentration in EMSA buffer (10 mM Tris-HCl (pH 7.5), 1 mM MgCl₂, 10 μM ZnCl₂, 1 mM DTT, 10 mM KCl, 0.5 mg/ml BSA). Serial dilutions of transcription factors were conducted in EMSA buffer to achieve the desired concentrations (ranging from 1 nM to 500 nM depending on the reaction). To test for affinity, 10 μl of diluted proteins of various concentrations were added to 10 μl of Cy-5 labeled DNA or nucleosomes. Reactions were then incubated at room temperature for 30 min in the dark followed by analysis in 5% nondenaturing polyacrylamide gels at 100 V for 75 min.

**Dissociation constants**. Apparent K_Ds were calculated from two separate EMSA binding curves per sample, each representing one independent experiment. Image analysis was conducted using image J[127]. The experimental data was analyzed using the 'non-linear regression' function with 'One site – Total' in GraphPad Prism (v8.0) software[14,15,128]. Bmax less than 1 and R² values between 0.8 and 0.99 were met to ensure actual fit of data[14,15,128]. K_Ds were either computed from the reduction in DNA or nucleosome bound fractions (designated total K_D) or based on the first appearance of a DNA or nucleosome bound complex (designated specific K_D) as in Ref.[14,15].

**RNA-seq and data analysis**. For RNA-seq, two biological replicates were generated for each treatment (mock or steroid) and timepoint (1, 6, or 24 h treatment). Root explants were treated with either 5 μM dexamethasone or mock solution for 24 h, 6 h, or 1 h before the end of the 5-day incubation on CIM plates. RNA from each sample (ca. 0.2 g) was purified using the RNeasy mini kit (Qiagen, 74104) after TRIzol (Invitrogen, 15596026) extraction[129]. RNA secondary structure was removed by a 5 min 65 °C incubation followed by immediate cool down. mRNA was selected with OligodT25 dynabeads (Invitrogen, 610-02). Reverse transcription was performed using the SSIII RT kit (Invitrogen, 18080-044) followed by end repair of cDNA using an enzyme mixture of T4 PNK and T4 DNA polymerase (Enzymatics Y9140-LC-L). After generating a 3′ A-overhang by Klenow HC (Enzymatics, P7010-HC-L), adapters were ligated with T4 DNA ligase (Enzymatics T4 DNA ligase (Rapid) #L603-HC-L, 600 U/μl). One-sided selection with SPRI-select beads (Beckman Coulter, B23317) was conducted before library amplifications with P5 and P7 index primers). Library quantification was performed with the NEBNext Library Quant Kit for Illumina (NEB, E7630). Single-end sequencing was conducted using NextSeq 550/500 High Output Kit v2 (Illumina, TG-160-2005) on the NextSeq500 platform (Illumina).

Quality control and filtering were identical to the ChIP-seq analysis. Reads were mapped using the STAR (v2.7.3)[130] mapping algorithm to the Araport11[131] annotation of the Arabidopsis genome (-outSAMmultNmax 1 -outMultimapperOrder Random -alignIntronMax 4350 -outFilterMultimapNmax 8 -outFilterMismatchNoverLmax 0.05)[130]. Specific read-coverage was assessed with HT-Seq (-stranded='no' -minaqual=30)[132]. For quality control, Spearman correlation coefficients of the reads in all protein coding and miRNA genes (Araport 11) of all biological replicates were compared using deepTools[114,115].

Pairwise differential expression analyses were performed by comparing pooled normalized read counts from dexamethasone- to mock-treated samples using default DESeq2 (v3.9)[133] parameters with normal shrinkage and adjusted[134] *p*-value cutoff of less than 0.01[133]. This yielded 54, 189, 2042 differentially

expressed genes 1, 6, and 24 h after dexamethasone relative to mock treatment, respectively.

**Peak annotation and dataset comparisons**. Significant LFY ChIP peaks were annotated to release 11 of Arabidopsis genome annotation (Araport11)[131]. Two rounds of annotation were performed. First, all peaks that were 3 kb upstream, or within genic regions were annotated to that gene. Second, orphan peaks were annotated to the nearest LFY dependent gene within 10 kb of the peak. LFY dependent genes are defined as genes that displayed rapid changes in gene expression after LFY-GR activation (this study and Ref. [57]). Plant GOSlim analyses were performed in AgriGO v2.0[135].

**Structural analysis of DNA binding domains**. Structures of linker histone H1 (PDB: 5NL0), LFY (PDB: 2VY1), FoxA2 (PDB: 5X07) and ARF1 (PDB: 4LDX) were visualized and aligned using PyMOL v2.3 (method=super, 5 cycles, cutoff = 2.0)[128].

**Statistical analysis and replication**. For all qPCR data, the Kolmogorov–Smirnov (K–S)[136] test was implemented to assess normal distribution of the data. Since all data were normally distributed, unpaired one-tailed $t$-tests were used to test whether changes in one specific direction were statistically significant and two-tailed $t$-tests were used to test changes in any direction. Error bars represent standard error of the mean (SEM) of at least two independent biological replicates. The hypergeometric test[137] was used to test whether two datasets significantly overlapped.

**Reporting Summary**. Further information on research design is available in the Nature Research Reporting Summary linked to this article.

## Data availability

The authors declare that the data supporting the findings of this study are available within the paper and its supplementary information files. The ChIP-seq, MNase-seq and RNA-seq datasets generated in this study are available at the GEO repository under accession number GSE141706. Source data are provided with this paper.

## Code availability

Scripts for peak to gene annotation: https://github.com/sklasfeld/ChIP_Annotation.

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

## Acknowledgements

We are grateful to Dr. Kim Gallagher and Dr. John Wagner for feedback on the manuscript. We thank Dr. E. Rhoades and K. McKibben (Department of Chemistry, University of Pennsylvania) for access to their HPLC machine and for support with anion exchange chromatography. We thank Dr. L. Strader (Washington University at St. Louis) for sharing protein purification protocols. We acknowledge the support by Dr. Greg Donahue's (Department of Cell and Molecular biology, SOM, University of Pennsylvania) with MNase-seq data analysis and thank members of the Wagner lab for feedback on experiments and the manuscript. This research was funded by NSF IOS grants 1557529 and 1905062.

## Author contributions

D.W., R.J., and S.K. conceived of the study. S.K. and R.J. conducted the bioinformatic analyses. R.J. and Y.Z. performed all wet lab experiments. J.X. contributed to genomic analyses, M.F.G. assisted with the nucleosomal EMSAs and A.K. contributed to LFY purification. S.-K.H. generated the gSWI3B-3xHA construct. D.W. wrote the paper with input from all authors.

## Competing interests

The authors declare no competing interests.
