## [Peer Review File · Nature Communications]

REVIEWER COMMENTS

Reviewer #1 (Remarks to the Author):

As usual, I disclose my identity: Rainer Melzer, University College Dublin

Pioneer transcription factors are of fundamental importance during plant and animal development because, in contrast to other transcription factors, they can also bind to nucleosomal DNA and initiate a change in chromatin state. This in turn facilitates binding of other transcription factors and hence developmental reprogramming.

However, despite their supposedly importance for plant development, few pioneer transcription factors have been identified in plants. In the present manuscript, Jin et al. present a plethora of in vitro and in vivo evidences suggesting that the master regulator LEAFY is a pioneer transcription factor. For example, the authors show that LFY binds to nucleosomal DNA in vitro and in vivo and is capable of displacing the H1 linker histone. The manuscript is well written and the results are of significant relevance for a broad range of molecular biologists and geneticists. I only have a couple of minor comments which are listed below.

Line 28: There is some evidence that SEP3 and AP1 act as pioneer transcription factors (Pajoro et al., 2014, Genome Biology) and although the evidence presented for LFY here is more comprehensive I'd avoid a statement like 'no bonafide pioneer transcription factor has been identified in this kingdom.'

Line 84: The link between the plant and mammalian NF-Y proteins and the evidence for their role as pioneer TFs is not entirely clear from the description provided.

Line 162: This and the subsequent paragraph are quite technical in their description and sometimes hard to follow.

Line 171 onward: Why was there no high MNase concentration mock control conducted?

Line 353: Here and elsewhere in the discussion: what exactly is meant with 'unlocking'? It might be good to have a more explicit mechanistic explanation here. Is it proposed that the histone-DNA interaction 'loosened' by LFY?

Figure 1 c: What is the unit for the dissociation constants? It might also make sense to provide the standard error for those measurements.

Figure 1c: NA should read ND (or ND in the legend should read NA)?

Supplementary Figure 1c: What is the explanation for the multiple protein-DNA complexes observed for LIM2?

Reviewer #2 (Remarks to the Author):

Jin et al explore whether LFY acts as a pioneer transcription factor. The concept of pioneer transcription factors have been defined from work in mammalian cells and during development. Although it seems likely that they would exist in plants, no bonafide pioneer TFs have yet been described. A paper from 2017 referred to LEC1 as a pioneer TF in the context of FLC reprogramming during embryogenesis, but none of the defining criteria for pioneer TFs were queried. Thus, this work represents an important advance in understanding the developmental regulation of transcription. The authors clearly lay out the criteria for pioneer TFs in the introduction and systematically test these through a variety of biochemical and in vivo approaches. The root explant LFY induction system is particularly powerful for the in vivo work as is their exploitation of LMI2, which they can use to establish that LFY functions distinctly from downstream factors. Together, the presented experiments and conclusions represent an important advance suited for publication in Nature Communications.

There are few points that need to be strengthened, given the data as currently presented, or the conclusions downplayed.

1. In the paragraph starting on line 110, the apparent dissociation constant of LFY binding to nucleosomal and naked DNA is referred to as being high. Please state the K_D in the text and what the known K_D range is for mammalian pioneer TFs. In comparison to values reported in Soufi et al (2014), the value for LFY is much higher, indicating a lower affinity.

2. Figure 3 is sequential ChIP to show that LFY binds a nucleosome (LFY IP then H3 IP) to further support the results in Figure 2. Sequential ChIP is a challenging experiment and I'm not confident in the authors' conclusions given the data as presented. For this experiment, chromatin was sheared to ~150 bp. From the methods it does not appear that the mononucleosome band was purified, which is critical to ensure LFY is binding a nucleosome. Smaller or larger fragments will also be present in chromatin sheared to an average size of 150 bp, potentially allowing the detection of LFY bound to non-nucleosomal DNA. Additionally, only two biological replicates were performed. I'd like to see more repetition of this experiment given the small fold change observed. Finally, the data are presented as a box plot. It is not appropriate to use a box plot for 2 data points. Each replicate should be shown with mean +/- SD based on the technical replicates. There is no way for the reader to determine how similar the biological replicates were or what the technical variation was within a biological replicate.

3. Figure 6 is an important figure for showing the consequences of LFY binding to nucleosomal DNA. There are two biological replicates for all experiments. The same comments about replicates and data presentation as for Figure 3 also apply here.

4. The last part of the results, Figure 7, appears more subjective than the rest of the data presented. The authors conclude that the structure of the LFY-DNA contact helix is highly similar to FoxA (line 289). By what quantitative metric? How is one to judge the images in Figure 7A?

5. The extensive discussion of ARF1 DNA contacts does not seem very relevant (lines 291-301). The fact that one TF binds DNA through B-sheets and another through alpha-helices does not mean those properties generally characterize pioneer TF and non-pioneer TFs.

6. The authors argue that they "uncovers striking similarities between plant and animal pioneer TFs". Because the definition of pioneer TFs comes from studies in animals and because only animal pioneer TFs have been characterized, it seems inevitable that plant pioneer TFs would have similar properties to animal pioneer TFs and that this is not a surprising result.

Minor corrections:

-Line 49: 147 base-pairs of DNA around octamer, not 150

-Line 334: missing word - "is delayed relative LFY activity"

-Line 164: incomplete sentence starting with "ChIP-Seq...."

-Line 288: reference should be to Figure 7?

-Sup Fig. 4 legend: part e is listed before part d

-Legend for Fig 2 is confusing. I at first took MNase 1 hour mock to mean a mock MNase treatment, not mock LFY induction. Consider rephrasing.

-If LFY ChIP-seq data in 2B and 2C are the same, only need to show it once. REVIEWER COMMENTS

Reviewer #1 (Remarks to the Author):

As usual, I disclose my identity: Rainer Melzer, University College Dublin

Pioneer transcription factors are of fundamental importance during plant and animal development because, in contrast to other transcription factors, they can also bind to nucleosomal DNA and initiate a change in chromatin state. This in turn facilitates binding of other transcription factors

and hence developmental reprogramming.

However, despite their supposedly importance for plant development, few pioneer transcription factors have been identified in plants. In the present manuscript, Jin et al. present a plethora of in vitro and in vivo evidences suggesting that the master regulator LEAFY is a pioneer transcription factor. For example, the authors show that LFY binds to nucleosomal DNA in vitro and in vivo and is capable of displacing the H1 linker histone. The manuscript is well written and the results are of significant relevance for a broad range of molecular biologists and geneticists. I only have a couple of minor comments which are listed below.

Line 28: There is some evidence that SEP3 and AP1 act as pioneer transcription factors (Pajoro et al., 2014, Genome Biology) and although the evidence presented for LFY here is more comprehensive I'd avoid a statement like 'no bonafide pioneer transcription factor has been identified in this kingdom.'

Line 84: The link between the plant and mammalian NF-Y proteins and the evidence for their role as pioneer TFs is not entirely clear from the description provided.

Line 162: This and the subsequent paragraph are quite technical in their description and sometimes hard to follow.

Line 171 onward: Why was there no high MNase concentration mock control conducted?

Line 353: Here and elsewhere in the discussion: what exactly is meant with 'unlocking'? It might be good to have a more explicit mechanistic explanation here. Is it proposed that the histone-DNA interaction 'loosened' by LFY?

Figure 1 c: What is the unit for the dissociation constants? It might also make sense to provide the standard error for those measurements.

Figure 1c: NA should read ND (or ND in the legend should read NA)?

Supplementary Figure 1c: What is the explanation for the multiple protein-DNA complexes observed for LIM2?

Reviewer #2 (Remarks to the Author):

Jin et al explore whether LFY acts as a pioneer transcription factor. The concept of pioneer transcription factors have been defined from work in mammalian cells and during development. Although it seems likely that they would exist in plants, no bonafide pioneer TFs have yet been described. A paper from 2017 referred to LEC1 as a pioneer TF in the context of FLC reprogramming during embryogenesis, but none of the defining criteria for pioneer TFs were queried. Thus, this work represents an important advance in understanding the developmental regulation of transcription. The authors clearly lay out the criteria for pioneer TFs in the introduction and systematically test these through a variety of biochemical and in vivo approaches. The root explant LFY induction system is particularly powerful for the in vivo work as is their exploitation of LIM2, which they can use to establish that LFY functions distinctly from downstream factors. Together, the presented experiments and conclusions represent an important advance suited for publication in Nature Communications.

There are few points that need to be strengthened, given the data as currently presented, or the conclusions downplayed.

1. In the paragraph starting on line 110, the apparent dissociation constant of LFY binding to nucleosomal and naked DNA is referred to as being high. Please state the KD in the text and what the known KD range is for mammalian pioneer TFs. In comparison to values reported in Soufi et al (2014), the value for LFY is much higher, indicating a lower affinity.

2. Figure 3 is sequential ChIP to show that LFY binds a nucleosome (LFY IP then H3 IP) to further support the results in Figure 2. Sequential ChIP is a challenging experiment and I'm not confident in the authors' conclusions given the data as presented. For this experiment, chromatin was sheared to ~150 bp. From the methods it does not appear that the mononucleosome band was

purified, which is critical to ensure LFY is binding a nucleosome. Smaller or larger fragments will also be present in chromatin sheared to an average size of 150 bp, potentially allowing the detection of LFY bound to non-nucleosomal DNA. Additionally, only two biological replicates were performed. I'd like to see more repetition of this experiment given the small fold change observed. Finally, the data are presented as a box plot. It is not appropriate to use a box plot for 2 data points. Each replicate should be shown with mean +/- SD based on the technical replicates. There is no way for the reader to determine how similar the biological replicates were or what the technical variation was within a biological replicate.

3. Figure 6 is an important figure for showing the consequences of LFY binding to nucleosomal DNA. There are two biological replicates for all experiments. The same comments about replicates and data presentation as for Figure 3 also apply here.

4. The last part of the results, Figure 7, appears more subjective than the rest of the data presented. The authors conclude that the structure of the LFY-DNA contact helix is highly similar to FoxA (line 289). By what quantitative metric? How is one to judge the images in Figure 7A?

5. The extensive discussion of ARF1 DNA contacts does not seem very relevant (lines 291-301). The fact that one TF binds DNA through B-sheets and another through alpha-helices does not mean those properties generally characterize pioneer TF and non-pioneer TFs.

6. The authors argue that they "uncover striking similarities between plant and animal pioneer TFs". Because the definition of pioneer TFs comes from studies in animals and because only animal pioneer TFs have been characterized, it seems inevitable that plant pioneer TFs would have similar properties to animal pioneer TFs and that this is not a surprising result.

Minor corrections:

-Line 49: 147 base-pairs of DNA around octamer, not 150

-Line 334: missing word - "is delayed relative LFY activity"

-Line 164: incomplete sentence starting with "ChIP-Seq..."

-Line 288: reference should be to Figure 7?

-Sup Fig. 4 legend: part e is listed before part d

-Legend for Fig 2 is confusing. I at first took MNase 1 hour mock to mean a mock MNase treatment, not mock LFY induction. Consider rephrasing.

-If LFY ChIP-seq data in 2B and 2C are the same, only need to show it once.

REVIEWER COMMENTS

Reviewer #1 (Remarks to the Author):

As usual, I disclose my identity: Rainer Melzer, University College Dublin

Pioneer transcription factors are of fundamental importance during plant and animal development because, in contrast to other transcription factors, they can also bind to nucleosomal DNA and initiate a change in chromatin state. This in turn facilitates binding of other transcription factors and hence developmental reprogramming.

However, despite their supposedly importance for plant development, few pioneer transcription factors have been identified in plants. In the present manuscript, Jin et al. present a plethora of in vitro and in vivo evidences suggesting that the master regulator LEAFY is a pioneer transcription factor. For example, the authors show that LFY binds to nucleosomal DNA in vitro and in vivo and is capable of displacing the H1 linker histone. The manuscript is well written and the results are of significant relevance for a broad range of molecular biologists and geneticists. I only have a couple of minor comments which are listed below.

Line 28: There is some evidence that SEP3 and AP1 act as pioneer transcription factors (Pajoro et al., 2014, Genome Biology) and although the evidence presented for LFY here is more comprehensive I'd avoid a statement like 'no bonafide pioneer transcription factor has been identified in this kingdom.'

Thank you for this comment, we have rephrased this sentence

Line 84: The link between the plant and mammalian NF-Y proteins and the evidence for their role as pioneer TFs is not entirely clear from the description provided.

We have restructured this paragraph to enhance clarity

Line 162: This and the subsequent paragraph are quite technical in their description and sometimes hard to follow.

We rewrote this paragraph in an attempt to make it less technical

Line 171 onward: Why was there no high MNase concentration mock control conducted?

We added these data to the current manuscript (we now show four MNase-seq datasets).

Line 353: Here and elsewhere in the discussion: what exactly is meant with 'unlocking'? It might be good to have a more explicit mechanistic explanation here. Is it proposed that the histone-DNA interaction 'loosened' by LFY?

Yes, it is proposed that LFY may directly or indirectly open the nucleosome containing its binding site to allow access of other transcription factors. LFY may directly loosen histone DNA interactions (some animal pioneer TFs seem to do this) or alternatively allow access to nucleosomal or nucleosome proximal DNA by linker histone displacement and chromatin remodeler recruitment.

Figure 1 c: What is the unit for the dissociation constants? It might also make sense to provide the standard error for those measurements.

We apologize for this omission, the unit is nM. We now compute and report two apparent KDs for each assay, total and specific, as described by Soufi et al. 2015 and by Fernandez Garcia et al. 2019.

Figure 1c: NA should read ND (or ND in the legend should read NA)?

Thank you for pointing out this inconsistency. Both legend and figure now report ND = not detectable.

Supplementary Figure 1c: What is the explanation for the multiple protein-DNA complexes observed for LIM2?

The reason for the multiple bands is not understood. To our knowledge there is only one LMI2 binding site in the 160 bp probe. We therefore hypothesize that LMI2 can form homodimers and higher order multimers.

Reviewer #2 (Remarks to the Author):

Jin et al explore whether LFY acts as a pioneer transcription factor. The concept of pioneer transcription factors have been defined from work in mammalian cells and during development. Although it seems likely that they would exist in plants, no bonafide pioneer TFs have yet been described. A paper from 2017 referred to LEC1 as a pioneer TF in the context of FLC reprogramming during embryogenesis, but none of the defining criteria for pioneer TFs were queried. Thus, this work represents an important advance in understanding the developmental regulation of transcription. The authors clearly lay out the criteria for pioneer TFs in the introduction and systematically test these through a variety of biochemical and in vivo approaches. The root explant LFY induction system is particularly powerful for the in vivo work as is their exploitation of LMI2, which they can use to establish that LFY functions distinctly from downstream factors. Together, the presented experiments and conclusions represent an important advance suited for publication in Nature Communications.

There are few points that need to be strengthened, given the data as currently presented, or the conclusions downplayed.

1. In the paragraph starting on line 110, the apparent dissociation constant of LFY binding to nucleosomal and naked DNA is referred to as being high. Please state the KD in the text and what the known KD range is for mammalian pioneer TFs. In comparison to values reported in Soufi et al (2014), the value for LFY is much higher, indicating a lower affinity.

We have updated our kD calculations to parallel those of Soufi et al. 2015 and Fernandez Garcia et al. 2019. We have also revised the text to include the apparent kDs identified in these prior studies for comparison.

2. Figure 3 is sequential ChIP to show that LFY binds a nucleosome (LFY IP then H3 IP) to further support the results in Figure 2. Sequential ChIP is a challenging experiment and I'm not confident in the authors' conclusions given the data as presented. For this experiment, chromatin was sheared to ~150 bp. From the methods it does not appear that the mononucleosome band was purified, which is critical to ensure LFY is binding a nucleosome. Smaller or larger fragments will also be present in chromatin sheared to an average size of 150 bp, potentially allowing the detection of LFY bound to non-nucleosomal DNA. Additionally, only two biological replicates were performed. I'd like to see more repetition of this experiment given the small fold change observed. Finally, the data are presented as a box plot. It is not appropriate to use a box plot for 2 data points. Each replicate should be shown with mean +/- SD based on the technical replicates. There is no way for the reader to determine how similar the biological replicates were or what the technical variation was within a biological replicate.

As the reviewer points out this is a very challenging experiment – since a lot of DNA is lost in the first ChIP so that the second ChIP is very tricky. As is, we use 2 g of tissue from two independent ChIP reactions for each final replicate; this corresponds to root explants generated from 200 plates of seedlings per replicate. It is therefore not feasible to isolate mono nucleosomes. We wish to point out that we are following the protocol by Iwafuchi -Choi and Zaret (2016). To address the reviewer's concern we have repeated the sequential ChIP experiment with three new biological replicates (new Fig. 3a), added an image to show the fragmentation by sonication (new Supplementary Fig. 5a) and included additional negative controls to rule out spurious histone association with the ChIP DNA (new Fig. 3a). These negative controls are comprised of LFY bound genes that either lack a nucleosome under the LFY peak summit (LSH2) or have weak nucleosomes under the LFY peak summit (ACR7) based on our LFY ChIP-seq and MNase-seq datasets. In addition, the original data are now provided as bargraphs in Supplementary Fig. 5.

3. Figure 6 is an important figure for showing the consequences of LFY binding to nucleosomal DNA. There are two biological replicates for all experiments. The same comments about replicates and data presentation as for Figure 3 also apply here.

As suggested by the reviewer we repeated the H1 ChIP qPCR, the SWI3B ChIP qPCR, and the FAIRE qPCR at two timepoints with two new biological replicates. The new data are of excellent quality (new Fig. 6) and fully replicate the initial findings (now shown in Supplementary Fig. 5).

4. The last part of the results, Figure 7, appears more subjective than the rest of the data presented. The authors conclude that the structure of the LFY-DNA contact helix is highly similar to FoxA (line 289). By what quantitative metric? How is one to judge the images in Figure 7A?

Thank you very much for this comment. We have rephrased the text to reflect that both the FoxA and the LFY DNA binding domains make shallow DNA contacts. We note tht FoxA, like LFY has a helix turn helix DNA contact region as part of its winged helix DNA binding domain.

5. The extensive discussion of ARF1 DNA contacts does not seem very relevant (lines 291-301). The fact that one TF binds DNA through B-sheets and another through alpha-helices does not mean those properties generally characterize pioneer TF and non-pioneer TFs.

We agree with this concern and have rephrased the description of the ARF1 DNA contacts to address the concerns by the reviewer.

6. The authors argue that they “uncover striking similarities between plant and animal pioneer TFs”. Because the definition of pioneer TFs comes from studies in animals and because only animal pioneer TFs have been characterized, it seems inevitable that plant pioneer TFs would have similar properties to animal pioneer TFs and that this is not a surprising result.

That is a good point. We have changed the statement to “uncover striking similarities between LFY and animal pioneer TFs”. These similarities go beyond the definition of a given pioneer transcription factor, which encompasses nucleosome binding in vitro and in vivo, as well as ability to reprogram cell fate. The additional similarities include features shared by some and not other animal transcription factors, such as linker histone displacement, SWI/SNF recruitment and delayed target locus opening.

Minor corrections:

-Line 49: 147 base-pairs of DNA around octamer, not 150 *corrected*

-Line 334: missing word – “is delayed relative LFY activity” *corrected*

-Line 164: incomplete sentence starting with “ChIP-Seq...” *corrected*

-Line 288: reference should be to Figure 7? *corrected*

-Sup Fig. 4 legend: part e is listed before part d *corrected*

-Legend for Fig 2 is confusing. I at first took MNase 1 hour mock to mean a mock MNase treatment, not mock LFY induction. Consider rephrasing. *Agreed – this was rephrased.*

-If LFY ChIP-seq data in 2B and 2C are the same, only need to show it once. *Since the LFY peaks are ranked based on the MNase signal strength the heatmaps are slightly different in each case.*

** See Nature Research’s author and referees’ website at www.nature.com/authors for information about policies, services and author benefits.

COVID 19 and impact on peer review

As a result of the significant disruption that is being caused by the COVID-19 pandemic we are very aware that many researchers will have difficulty in meeting the timelines associated with our peer review process during normal times. Please do let us know if you need additional time.

Our systems will continue to remind you of the original timelines but we intend to be highly flexible at this time.

This email has been sent through the Springer Nature Tracking System NY-610A-NPG&MTS

Confidentiality Statement:

This e-mail is confidential and subject to copyright. Any unauthorised use or disclosure of its contents is prohibited. If you have received this email in error please notify our Manuscript Tracking System Helpdesk team at <http://platformsupport.nature.com>.

Details of the confidentiality and pre-publicity policy may be found here <http://www.nature.com/authors/policies/confidentiality.html>

Privacy Policy | Update Profile

DISCLAIMER: This e-mail is confidential and should not be used by anyone who is not the original intended recipient. If you have received this e-mail in error please inform the sender and delete it from your mailbox or any other storage mechanism. Springer Nature Limited does not accept liability for any statements made which are clearly the sender's own and not expressly made on behalf of Springer Nature Ltd or one of their agents.

Please note that Springer Nature Limited and their agents and affiliates do not accept any responsibility for viruses or malware that may be contained in this e-mail or its attachments and it is your responsibility to scan the e-mail and attachments (if any).

Springer Nature Ltd. Registered office: The Campus, 4 Crinan Street, London, N1 9XW.

Registered Number: 00785998 England.

REVIEWERS' COMMENTS

Reviewer #1 (Remarks to the Author):

It would be nice to briefly address the multiple bands in the LIM2 gel shifts also in the legend of supplementary figure 1 (as far as I can see they have been only addressed in the response letter).

Other than that, I have no additional comments concerning the manuscript. It would have been helpful to have line numbers also in the revised version and to have a more extensive response letter, this would have made the re-review easier.

Rainer Melzer, University College Dublin

Reviewer #2 (Remarks to the Author):

I appreciate the efforts the authors have made in revising the manuscript. All of my comments have been satisfactorily addressed. This is an excellent study.

Minor typos:

On p. 4, end of intro, change "call fate" to "cell fate"

On p. 6, first paragraph, "shows very high affinity" should be "shows higher affinity"

REVIEWERS' COMMENTS

Reviewer #1 (Remarks to the Author):

It would be nice to briefly address address the multiple bands in the LIM2 gel shifts also in the legend of supplementary figure 1 (as far as I can see they have been only addressed in the response letter).

Other than that, I have no additional comments concerning the manuscript. It would have been helpful to have line numbers also in the revised version and to have a more extensive response letter, this would have made the re-review easier.

Thank you for these suggestions. We have added a sentence discussing the multiple shifted bands in the LMI2 EMSA to the legend of Supplementary Fig. 1

Rainer Melzer, University College Dublin

Reviewer #2 (Remarks to the Author):

I appreciate the efforts the authors have made in revising the manuscript. All of my comments have been satisfactorily addressed. This is an excellent study.

Minor typos:

On p. 4, end of intro, change “call fate” to “cell fate”

On p. 6, first paragraph, “shows very high affinity” should be “shows higher affinity”

Thank you for pointing out these typos, we corrected them.